# A novel region in the Ca$_V$2.1 $\alpha_1$ subunit C-terminus regulates fast synaptic vesicle fusion and vesicle docking at the mammalian presynaptic active zone

Matthias Lübbert[1†], R Oliver Goral[1,2†], Rachel Satterfield[1], Travis Putzke[1], Arn MJM van den Maagdenberg[3], Naomi Kamasawa[4], Samuel M Young Jr[1,2,5,6,7*‡]

[1]Research Group Molecular Mechanisms of Synaptic Function, Max Planck Florida Institute for Neuroscience, Jupiter, United States; [2]Department of Anatomy and Cell Biology, University of Iowa, Iowa City, United States; [3]Departments of Human Genetics and Neurology, Leiden University Medical Center, Leiden, Netherlands; [4]Max Planck Florida Electron Microscopy Core, Max Planck Florida Institute for Neuroscience, Jupiter, United States; [5]Department of Otolaryngology, University of Iowa, Iowa City, United States; [6]Iowa Neuroscience Institute, University of Iowa, Iowa City, United States; [7]Aging Mind Brain Initiative, University of Iowa, Iowa City, United States

*For correspondence: samuel-m-young@uiowa.edu

†These authors contributed equally to this work

Present address: ‡Anatomy and Cell Biology, University of Iowa, Iowa City, United States

Competing interests: The authors declare that no competing interests exist.

**Abstract** In central nervous system (CNS) synapses, action potential-evoked neurotransmitter release is principally mediated by Ca$_V$2.1 calcium channels (Ca$_V$2.1) and is highly dependent on the physical distance between Ca$_V$2.1 and synaptic vesicles (coupling). Although various active zone proteins are proposed to control coupling and abundance of Ca$_V$2.1 through direct interactions with the Ca$_V$2.1 $\alpha$1 subunit C-terminus at the active zone, the role of these interaction partners is controversial. To define the intrinsic motifs that regulate coupling, we expressed mutant Ca$_V$2.1 $\alpha_1$ subunits on a Ca$_V$2.1 null background at the calyx of Held presynaptic terminal. Our results identified a region that directly controlled fast synaptic vesicle release and vesicle docking at the active zone independent of Ca$_V$2.1 abundance. In addition, proposed individual direct interactions with active zone proteins are insufficient for Ca$_V$2.1 abundance and coupling. Therefore, our work advances our molecular understanding of Ca$_V$2.1 regulation of neurotransmitter release in mammalian CNS synapses.

## Introduction

A critical determinant in regulating synaptic vesicle (SV) release probability and kinetics is coupling, the physical distance of SVs and voltage-gated calcium channels (VGCCs) at the presynaptic terminal (*Neher and Sakaba, 2008*). Differences in coupling distances between Ca$_V$2 VGCCs subtypes underpin the differences in Ca$_V$2 VGCC subtype effectiveness in eliciting AP evoked release and define the SV release mode in response to APs (*Eggermann et al., 2011*). They are: nanodomain, a few tightly coupled VGCCs (<30 nm), and microdomain, many loosely coupled VGCCs (~100 nm) trigger SV release (*Baur et al., 2015*; *Eggermann et al., 2011*; *Fedchyshyn and Wang, 2005*). In the majority of central nervous system synapses, Ca$_V$2.1 VGCCs (Ca$_V$2.1) are the principal Ca$_V$ subtype that supports AP mediated neurotransmitter release, and Ca$_V$2.1 channels are thought to exist in closest proximity to SVs compared to other Ca$_V$ subtypes (*Eggermann et al., 2011*). The Ca$_V$2.1 $\alpha_1$ subunit cytoplasmic C-terminus is mutated in a class of Ca$_V$2 channelopathies (*Pietrobon, 2010*) and

**eLife digest** The points of contact between nerve cells are called synapses, and nerve cells communicate across synapses via chemicals known as neurotransmitters. These chemical messengers are initially stored within bubble-like packages called synaptic vesicles that are released after they fuse with the membrane of the nerve cell at a specialized site referred to as the "active zone".

Calcium ions are one of the major factors that lead to the release of synaptic vesicles. Ion channel proteins in the membrane of the nerve cell control the flow of calcium ions into the cell. There are often many different ion channels at a synapse, but one type called $Ca_V2.1$ most effectively triggers the release of synaptic vesicles when a nerve impulse reaches the synapse. Various proteins at the active zone can bind directly to parts of the $Ca_V2.1$ channel that are identified by a short sequence of amino acids – the building blocks of all proteins. Several researchers have proposed that the interactions with some of these short sequences, which are also known as motifs, control how much of this ion channel is in the synapse and how it interacts with synaptic vesicles to regulate the release of neurotransmitters. However, other researchers do not agree with this proposed explanation.

Lübbert, Goral et al. set out to determine which parts in a specific part of the $Ca_V2.1$ channel (called the "α1 subunit C-terminus") are critical for its interaction with synaptic vesicles. The experiments revealed a new motif that regulates how many synaptic vesicles could be released in response to electrical impulses travelling along nerve cells from mice. The same motif also regulates the total number of synaptic vesicles at the active zone.

Lübbert, Goral et al. went on to show that binding to known active proteins at most played a minor role in controlling the abundance of the $Ca_V2.1$ channels and how close they were to the synaptic vesicles. As such, these findings counter prevailing views of the roles of certain motifs in the α1 subunit of the $Ca_V2.1$ channel. Thus, it may be necessary to re-think how the $Ca_V2.1$ channel regulates the release of synaptic vesicles.

Ion channels are vital to the activity of all nerve cells, and working out how the numbers and organization of $Ca_V2.1$ and related ion channels are regulated will be fundamental to understanding how information is encoded in brain. In addition, problems with these kinds of ion channel may result in disorders such as migraines and epilepsy. Therefore, the new findings may help to guide further studies investigating possible ways to treat these disorders.

contains many motifs implicated to directly interact with key active zone (AZ) proteins to control $Ca_V2.1$ coupling and abundance in the presynaptic terminal (*Simms and Zamponi, 2014*). Nevertheless, the necessity and mechanism of action of these motifs are highly controversial due to disparate results from different model systems and from knockout mouse models of AZ proteins (*Acuna et al., 2015*; *Atasoy et al., 2007*; *Butz et al., 1998*; *Cao et al., 2004*; *Das, 2016*; *Davydova et al., 2014*; *Ho et al., 2006*; *Hu et al., 2005*; *Kaeser et al., 2011*; *Wong et al., 2014*; *Wong and Stanley, 2010*). In addition, it is unclear whether the mechanisms that control coupling and abundance are interrelated or separable.

To address these questions, we utilized the calyx of Held/Medial Nucleus of the Trapezoid Body (MNTB) synapse, a large glutamatergic axosomatic synapse, in which: (1) individual AZs ultrastructure and (2) $Ca_V2$ subtype abundance and proximity to SVs controlling SV release at the calyx of Held is similar to many other synapses (*Borst and Soria van Hoeve, 2012*). Furthermore, due to its unparalleled experimental accessibility, molecular manipulations can be made exclusively in the presynaptic terminals (*Wimmer et al., 2004*; *Young and Neher, 2009*), and presynaptic $Ca^{2+}$ currents can be recorded and correlated with synaptic vesicle release rates (*Neher and Sakaba, 2001b*), which allows for well-controlled measurements not achievable in other model systems. By directly manipulating the $Ca_V2.1$ $\alpha_1$ subunit in a native neuronal circuit, we were able to overcome the previous limitations in prior studies (*Acuna et al., 2015*; *Atasoy et al., 2007*; *Butz et al., 1998*; *Cao et al., 2004*; *Davydova et al., 2014*; *Ho et al., 2006*; *Hu et al., 2005*; *Kaeser et al., 2011*; *Wong et al., 2014*; *Wong and Stanley, 2010*). Thus, we were able to identify a novel intrinsic motif in the C-terminus that regulates coupling and demonstrate that coupling and abundance are separable. Finally, we found that this novel C-terminal region in the $Ca_V2.1$ $\alpha_1$ subunit also regulates SV

docking at the AZ. Therefore, our work provides new molecular insights into $Ca_V2.1$ $\alpha_1$ subunit regulation of SV release from presynaptic terminals in CNS synapses.

## Results

### Genetic manipulation of $Ca_V2.1$ channels at the calyx

To manipulate $Ca_V2.1$ at the calyx, Helper-Dependent Adenoviral vectors (HdAd) (*Palmer and Ng, 2005*) were utilized in conjunction with a *Cacna1a* conditional knock-out (CKO) mouse line (*Todorov et al., 2006*). HdAds can package large amounts of foreign DNA, which is critical as the $Ca_V2.1$ $\alpha_1$ subunit cDNA is larger than commonly used viral vectors (*Lentz et al., 2012*). To modify $Ca_V2.1$ expression at the calyx, we used stereotactic surgery to deliver our HdAd viral vectors expressing Cre recombinase (HdAd Cre) to create a *Cacna1a* null background ($Ca_V2.1^{-/-}$) and the

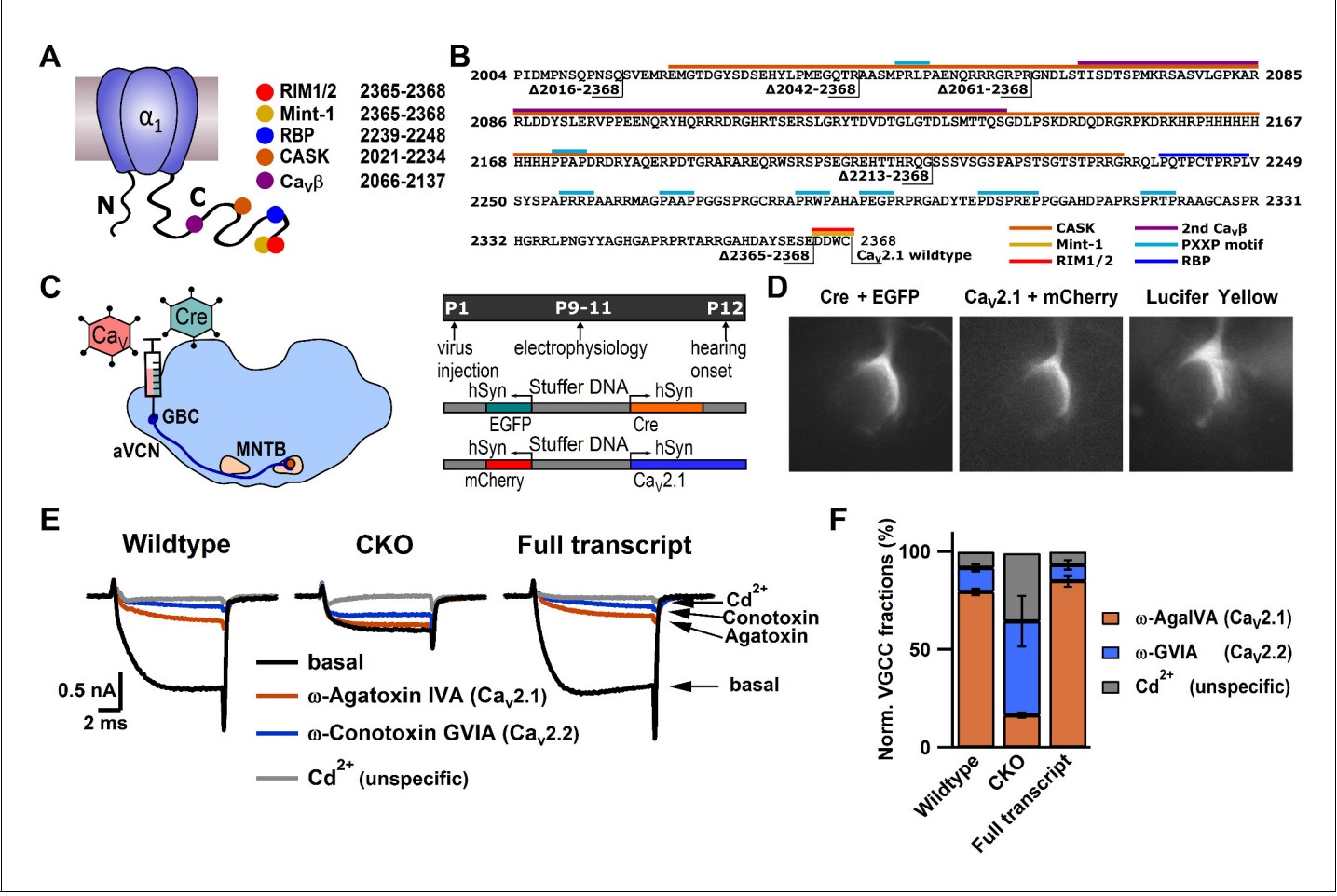

**Figure 1.** $Ca_V2.1$ can be selectively ablated and functionally rescued at the calyx of Held. (**A**) Cartoon depicting $Ca_V2.1$ $\alpha_1$ subunit distal C-terminal interaction partners. (**B**) Amino acid sequence of the distal Cav2.1 C-terminus indicating interaction sites and truncation mutants. (**C**) left: Schematic view of stereotactic surgery to inject/coinject HdAd vectors expressing Cre + eGFP and $Ca_V2.1$ constructs + mCherry into the aVCN at age P1. Right: top: Experimental timeline from virus injection at P1 to electrophysiological recordings at P9-P11 prior to the onset of hearing (P12). Middle and bottom: schematic view of the viral constructs used, expressing either Cre + eGFP or $Ca_V2.1$ constructs + mCherry, respectively, driven by individual promotors. (**D**) Calyx of Held terminals transduced with Cre + eGFP (top) and $Ca_V2.1$ + mCherry (middle). eGFP and mCherry signals overlap with those of a calyx of Held loaded with Lucifer Yellow via a patch pipette (bottom). (**E**) Pharmacological isolation of presynaptic $Ca_V2$ isoforms in wildtype, CKO and $Ca_V2.1$ full transcript rescue calyxes. Traces in absence of any blockers (black), after blocking $Ca_V2.1$ fraction with 200 nM ω-AgaIVA (brown), after blocking $Ca_V2.2$ fraction with 2 μM ω-GVIA (blue) and after blocking all $Ca_V2$ channels with 50 μM $Cd^{2+}$ (gray). (**F**) Relative $Ca_V2$ current fractions in wildtype, CKO and $Ca_V2.1$ full transcript rescue calyxes (n = 3 for each condition).

full transcript of Ca$_V$2.1 $\alpha_1$ subunit, (HdAd Ca$_V$2.1 FT) into the cochlear nucleus (*Chen et al., 2013*) (*Figure 1*). Ca$_V$2.1 full transcript (FT) is the longest Ca$_V$2.1 $\alpha_1$ subunit cDNA (*Mus musculus* NP_031604.3). By testing Ca$^{2+}$ current sensitivity to Ca$_V$2 subtype-specific blockers ω-Agatoxin IVA (Aga, Ca$_V$2.1-selective) and ω-Conotoxin GVIA (Cono, Ca$_V$2.2-selective), we confirmed that we could ablate Ca$_V$2.1 and subsequently rescue Ca$_V$2.1 abundance (*Figure 1*).

## Active zone protein binding sites in the Ca$_V$2.1 $\alpha_1$ subunit C-terminus are dispensable for Ca$_V$2.1 abundance in the presynaptic terminal

Since we could manipulate Ca$_V$2.1 at the calyx, we tested whether various previously proposed direct binding sites are necessary for regulating Ca$_V$2.1 localization and abundance at the presynaptic membrane. This includes binding sites for RIM1/2 (*Kaeser et al., 2011*), MINT1 (*Maximov et al., 1999*), Rim Binding Proteins (RBP) (*Hibino et al., 2002*), and CASK proteins (*Maximov et al., 1999*), a secondary Ca$_V$β4 interaction site (*Walker et al., 1998*) as well as PXXP motifs (*Davydova et al., 2014*). To do so we generated HdAd vectors with mutations in Ca$_V$2.1 $\alpha_1$ subunits in which we deleted these interaction sites (*Figures 1A–B*, *2* and *3A*). We expressed them at the Ca$_V$2.1$^{-/-}$ calyx and carried out whole-cell patch clamp recordings of the presynaptic Ca$^{2+}$ currents (*Figure 2—figure supplement 2 and 1* and *Table 1*); Ca$_V$2.1Δ2365–2368 deletes the $\alpha_1$ subunit DDWC motif that is implicated to bind directly to RIM1/2 and MINT (*Kaeser et al., 2011*). Ca$_V$2.1Δ2213–2368 corresponds to a Ca$_V$2.1 $\alpha_1$ subunit splice variant which removes the RIM1/2, MINT1, RBP, and part of the CASK binding site and majority of PXXP motifs (*Soong et al., 2002*). Ca$_V$2.1Δ2016–2368 removes the complete CASK binding site, a proposed secondary Ca$_V$β4 interaction site, and two remaining PXXP motifs in the $\alpha_1$ subunit (*Figure 1*, *Figure 2—figure supplement 1*). Analysis of the Ca$^{2+}$ current as a function of voltage (I(V)) and tail currents revealed that expression of mutants lacking motifs located within the last 350 amino acids revealed no significant difference in Ca$^{2+}$ current amplitudes or voltage dependent activation compared to Ca$_V$2.1 FT rescue (*Figure 2*, *Figure 2—figure supplement 1* and *Table 1*). Although there appeared to be a slight reduction in maximal Ca$^{2+}$ current amplitudes compared to FT rescue, there was no statistically significant difference among mutants and control. Thus, the MINT1, RIM1/2, RBP, CASK proteins and the secondary Ca$_V$β4 binding sites within the Ca$_V$2.1 $\alpha_1$ subunit C-terminus are not necessary for Ca$_V$2.1 localization to the presynaptic membrane.

## A novel C-terminal region in the Ca$_V$2.1 $\alpha_1$ subunit is required SV to Ca$_V$2.1 channel coupling and regulates fast vesicle fusion and RRP size

To determine the intrinsic motif(s) involved in the regulation of coupling, we performed paired whole cell voltage clamp recordings on the pre- and postsynaptic compartments of the calyx of Held/MNTB synapse with these deletion mutants (*Neher and Sakaba, 2001a*, *2001b*). For finer mapping we generated two additional deletion constructs (*Figure 3A*). Ca$_V$2.1Δ2061–2368 deletes up to the secondary Ca$_V$β4 interaction in the $\alpha_1$ subunit and Ca$_V$2.1Δ2042–2368 deletes an additional arginine rich stretch in the Ca$_V$2.1 $\alpha_1$ subunit, not found in Ca$_V$2.2 and Ca$_V$2.3 and the final shared PXPP motif. Conotoxin was included to block possible Ca$_V$2.2 channel contributions. First we applied either a 3 ms step depolarization pulse (*Figure 3—figure supplement 1A*) to the calyx which selectively depletes SVs within ~50–80 nm of Ca$_V$2 VGCCs (*Chen et al., 2015*) which participate in synchronous transmitter release (fast pool) (*Chen et al., 2015*; *Lee et al., 2012*). The fast pool is the relevant SV pool that supports AP-mediated release and thus considered the readily-releasable pool (RRP) (*Figures 3–4*, *Figure 3—figure supplement 1* and *Table 2*) (*Sakaba, 2006*). Then we applied a 30 ms step depolarization (*Figure 3—figure supplement 1*) which measures the entire pool of fusion competent SVs, all within ~200 nm of Ca$_V$2 and considered the total releasable pool (*Chen et al., 2015*; *Lee et al., 2012*) (*Figures 3–4*, *Figure 3—figure supplement 1* and *Table 2*). To validate our approach we compared the effects of SV release between calyces expressing Cre + Ca$_V$2.1 FT construct and wild-type calyces. We found no differences in SV release between the Ca$_V$2.1 FT and wild-type calyces (*Figure 3*, *Figure 3—figure supplement 1* and *Table 2*), indicating that exogenous expression of the Ca$_V$2.1 $\alpha_1$ subunit did not alter calyx/MNTB synaptic transmission.

In response to 3 ms and 30 ms presynaptic depolarizations, we found no difference in the presynaptic Ca$^{2+}$ currents in all mutants, thus confirming our results depicted in *Figure 2* (*Figure 3* and *Table 2*). Since the 3 ms peak EPSC amplitude directly correlates to those SVs that are tightly

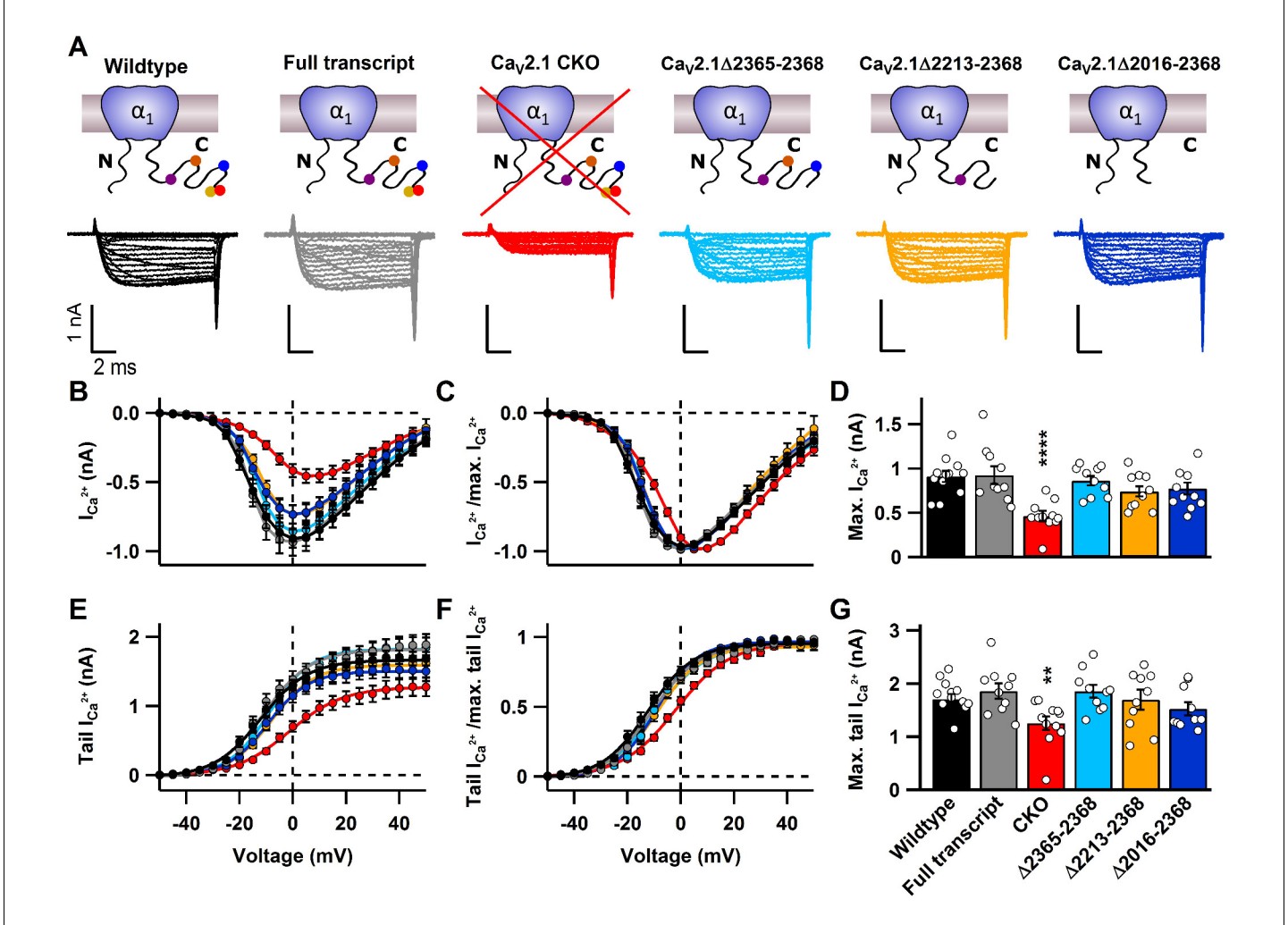

**Figure 2.** C-terminal deletions in $Ca_V2.1$ do not affect $Ca_V2$ abundance at the presynaptic terminal. (A) Cartoons depicting $Ca_V2.1$ full transcript or mutants (top) with corresponding exemplary $Ca^{2+}$ currents (bottom) triggered by 10 ms voltage steps from d -50 mV to 50 mV in 5mV steps. (B–C) Current-voltage relationship of absolute $Ca^{2+}$ currents (B) and normalized current-voltage relationships (I/$I_{max}$; C). (D) Mean absolute $Ca^{2+}$ currents. (E–F) Absolute tail currents (E) and normalized (I/$I_{max}$; F) tail currents as a function of voltage. (G) Mean tail $Ca^{2+}$ currents at +40 mV. For $Ca_V2.1$ $\alpha_1$ CKO (n = 11), wildtype (n = 12), $Ca_V2.1$ $\alpha_1$ full transcript (n = 10), $\Delta2365$–2368 (n = 10), $\Delta2213$–2368 (n = 10) and $\Delta2016$–2368 (n = 10). All data are depicted as mean ± SEM. Detailed values can be derived from *Table 1*.

The following figure supplement is available for figure 2:

**Figure supplement 1.** $Ca_V2.1$ rescue after CKO does not affect biophysical properties of the $Ca^{2+}$ current at the calyx of Held.

coupled to $Ca_V2$ channels at the P9-11 calyx (*Chen et al., 2015*; *Lee et al., 2012*), we measured the peak 3 ms EPSC peak amplitudes in all our deletion mutants. Thus, if these intrinsic motifs were essential for SV to $Ca_V2.1$ coupling, we should see a dramatic reduction in the 3 ms peak EPSC amplitude, and if they were not essential there should be no change. Analysis of the 3 ms peak EPSC amplitudes revealed no change in the peak amplitudes with deletions from amino acid 2265 and beyond ($Ca_V2.1\Delta2365$–2368, $Ca_V2.1\Delta2213$–2368 and $Ca_V2.1\Delta2061$–2368), when compared to control ($Ca_V2.1$ FT; *Figure 3* and *Table 2*). However, we saw a dramatic reduction in the EPSC amplitudes with $Ca_V2.1\Delta2042$–2368 and $Ca_V2.1\Delta2016$–2368 (FT: 8.54 ± 0.9 nA; $\Delta2042$–2368: 3.38 ± 0.89 nA (p<0.01); $\Delta2016$–2368: 2.47 ± 0.89 nA (p<0.001); *Figure 3* and *Table 2*). In addition, only the $Ca_V2.1\Delta2042$–2368 and $Ca_V2.1\Delta2016$–2368 mutants showed a significant slowdown in the 10–90

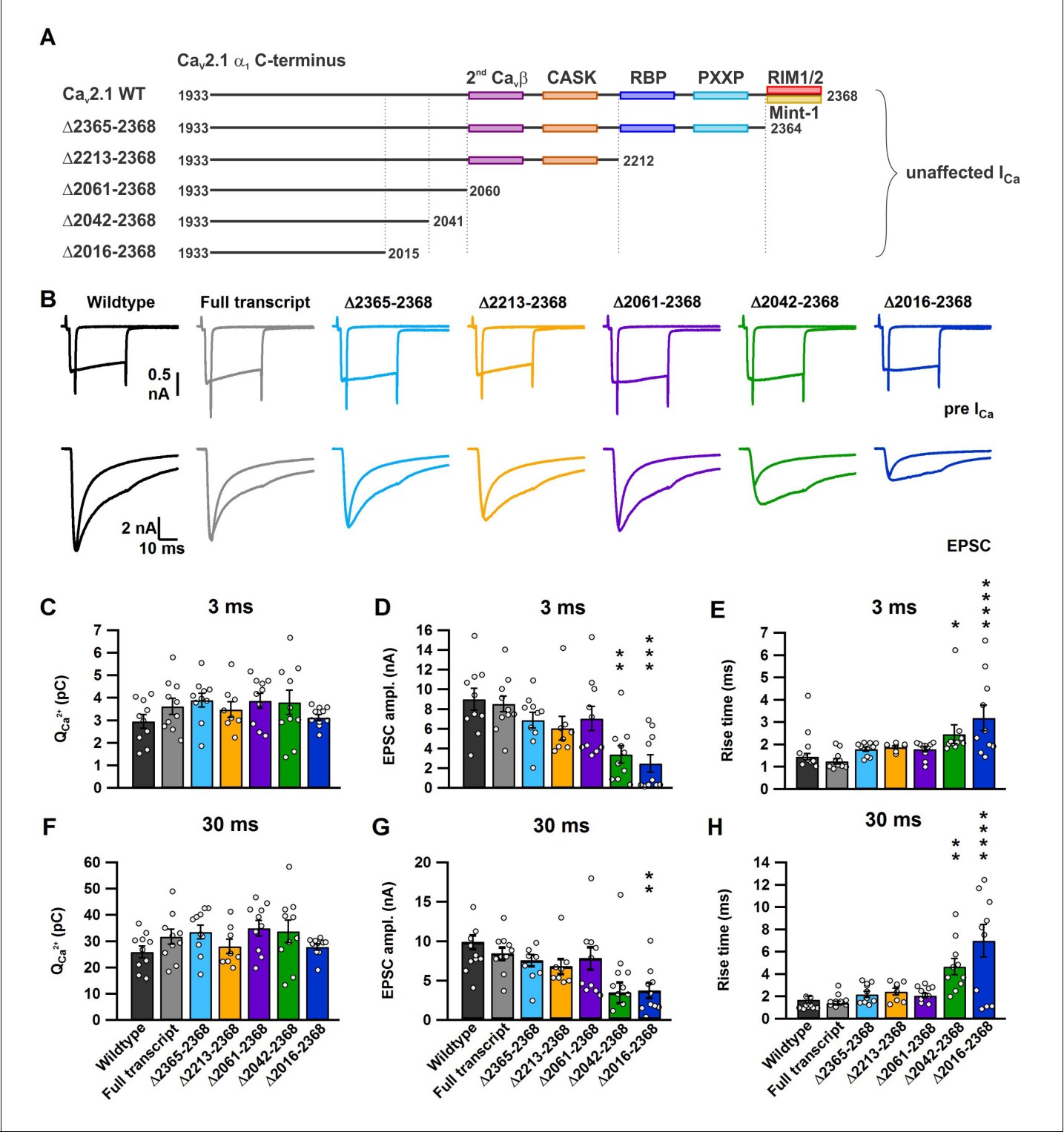

**Figure 3.** A novel role for a C-terminal region between amino acids 2042 and 2061 that regulates fast release independent of Ca$_V$2.1 abundance. (**A**) Cartoon depicting truncated regions in our Ca$_V$2.1 $\alpha_1$ deletion mutants including the binding sites for Ca$_V$$\beta$4, CASK, RBP, PXXP, RIM1/2 and Mint-1 along with the effects of C-terminal truncations on I$_{Ca}$. (**B**) Averaged traces of RRP and total releasable pool measurements from mice expressing Cre + full transcript Ca$_V$2.1 rescue (grey), $\Delta$2365–2368 (cyan), $\Delta$2213–2368 (yellow), $\Delta$2061–2368 (purple), $\Delta$2042–2368 (green) or $\Delta$2016–2368 (blue). I$_{Ca}$ (top) and EPSCs (bottom) triggered by 3 ms and 30 ms pulses, plotted on top of each other (n = 10 for each group, except for $\Delta$2212–2368: n = 8). (**C–H**)
*Figure 3 continued on next page*

Figure 3 continued

Quantification of $I_{Ca}$ charge (3 ms: **C**; and 30 ms: **F**), max. EPSC amplitudes (3 ms: **D**; 30 ms: **G**) and the 10–90% rise of the EPSCs (3 ms: **E**; 30 ms: **H**). All data are depicted as mean ± SEM. Detailed values can be derived from *Table 2*.

The following figure supplement is available for figure 3:

**Figure supplement 1.** $Ca_V2.1$ Full transcript rescue does not affect synaptic transmission at the calyx of Held/MNTB synapse.

rise time compared to FT and a significant increases in the synaptic delay time (*Figure 2* and *Table 2*).

In response to the 30 ms step pulse, we found no significant change in the 10–90 rise time or EPSC amplitudes with $Ca_V2.1\Delta2365–2368$, $Ca_V2.1\Delta2213–2368$ and $Ca_V2.1\Delta2061–2368$ compared to control (*Figure 2*, *Table 2*). It is important to note that unlike the 3 ms peak EPSC amplitude, the 30 ms 10–90 peak EPSC rise time is an inaccurate measure of coupling of all SVs in the total releasable pool, as the 30 ms peak amplitude does not accurately measure the total releasable pool size (*Chen et al., 2015*; *Lee et al., 2012*). We found a significant increase in the 10–90 rise time with $Ca_V2.1\Delta2042–2368$ and $Ca_V2.1\Delta2016–2368$ (FT: 1.41 ± 0.18 ms; $\Delta2042–2368$: 4.67 ± 0.72 ms (p<0.01); $\Delta2016–2368$: 6.99 ± 1.45 ms (p<0.0001)), with reduced EPSC amplitudes (*Figure 3* and *Table 2*). In all cases, there was no difference between $Ca_V2.1\Delta2042–2368$ and $Ca_V2.1\Delta2016–2368$ indicating that the further deletion did not lead to more severe reductions in the ESPC amplitude or 10–90 rise time. Although, there appeared to be a slight slowing in the 10–90 rise time with $Ca_V2.1\Delta2365–2368$, $Ca_V2.1\Delta2213–2368$ and $Ca_V2.1\Delta2061–2368$ compared to control in both the 3 ms and 30 ms EPSC (*Figure 3*, *Table 2*), this change was not statistically significant and was very minor compared to the dramatic deceleration in in the 10–90 rise times found in $Ca_V2.1\Delta2042–2368$ and $Ca_V2.1\Delta2016–2368$. Based on these results, we conclude that a novel C-terminal region including at least the amino acids 2042–2061 is critical for fast release.

## A novel C-terminal region in $Ca_V2.1$ $\alpha_1$ subunit regulates the total releasable pool size

To understand how the $Ca_V2.1\Delta2042–2368$ and $Ca_V2.1\Delta2016–2368$ truncations impacted the size and release kinetics of the fast pool (AP-evoked release) and the total releasable pool (*Figure 3* and *Figure 3—figure supplement 1*), we used a deconvolution analysis routine to calculate release rates (*Neher and Sakaba, 2001a*, *2001b*). We found that both $Ca_V2.1\Delta2042–2368$ and $Ca_V2.1\Delta2016–2368$ lead to a dramatic reduction in peak vesicle release rates (*Figure 4*) with a significant increase in the delayed release (slow pool component) and slower time to peak EPSC release rates compared to control (*Figure 4*). Integration of the release rates for both the 3 ms and 30 ms pulses revealed a dramatic reduction in both mutants of both the fast pool and the total releasable pool. (RRP: FT: 1505 ± 245 SVs; $\Delta2042–2368$: 430 ± 116 SVs (p<0.001); $\Delta2016–2368$: 357 ± 134 SVs (p<0.001); total releasable pool: FT: 2152 ± 263 SVs; $\Delta2042–2368$: 1292 ± 221 SVs (p<0.05); $\Delta2016–2368$: 1061 ± 258 SVs (p<0.01)) (*Figure 4*). To test how the kinetics of release were affected by the fast component of release, the cumulative release rates were normalized to their respective total number of vesicles released during the 30 ms depolarizing pulse (*Figure 4D*). This clearly demonstrates that both mutants had a significantly decreased fast component. In all cases, there were no differences between $Ca_V2.1\Delta2042–2368$ and $Ca_V2.1\Delta2016–2368$. Comparison of the ratio of the RPP size to the total releasable pool size revealed a significant reduction in the contribution of the RRP to the total releasable pool size in the mutants. (*Figure 4G*)(*Table 3*). Thus, based on our results we can conclude that the region between 2016 and 2042 is essential for both regulating the total number of releasable vesicles, as well as the relative contributions of fast and slow SV pool components.

## A novel C-terminal region in $Ca_V2.1$ $\alpha_1$ subunit regulates SV Docking at the active zone

Since docked synaptic vesicles at the AZ are the morphological correlates of the RRP (*Schikorski and Stevens, 2001*), we next assessed how $Ca_V2.1\Delta2042–2368$ and $Ca_V2.1\Delta2016–2368$ affected presynaptic ultrastructure. To do so, we acquired and analyzed electron microscopy (EM)

**Table 1.** Electrophysiological parameters of IV relations of Ca$^{2+}$ currents.

| Parameter | Mean ± SEM (n) | OW-ANOVA Dunnett's Test |
|---|---|---|
| **Max. Ca$^{2+}$ current amplitude I$_{max}$ (pA)** | | |
| Wild type | 911 ± 63 (12) | p=0.9998 (n.s.) |
| Full transcript | 925 ± 99 (10) | control group |
| CKO | 464 ± 59 (11) | p<0.0001 (****) |
| △2365–2368 | 863 ± 53 (10) | p=0.9513 (n.s.) |
| △2213–2368 | 741 ± 56 (10) | p=0.2261 (n.s.) |
| △2016–2368 | 744 ± 67 (10) | p=0.2383 (n.s.) |
| **Membrane capacitance C$_{slow}$ (pF)** | | |
| Wild type | 20.9 ± 1.6 (12) | p=0.5071 (n.s.) |
| Full transcript | 18.2 ± 1.2 (10) | control group |
| CKO | 18.1 ± 1.8 (11) | p>0.9999 (n.s.) |
| △2365–2368 | 16.9 ± 1.1 (10) | p=0.9593 (n.s.) |
| △2213–2368 | 19.8 ± 1.4 (10) | p=0.9933 (n.s.) |
| △2016–2368 | 17.2 ± 1.8 (10) | p=0.8986 (n.s.) |
| **IV fit: Half-maximal activation voltage V$_m$ (mV)** | | |
| Wild type | −24.6 ± 1.3 (12) | p=0.9986 (n.s.) |
| Full transcript | −25.1 ± 1.3 (10) | control group |
| CKO | −22.3 ± 1.3 (11) | p=0.3604 (n.s.) |
| △2365–2368 | −23.1 ± 1.1 (10) | p=0.6831 (n.s.) |
| △2213–2368 | −23.8 ± 1.5 (10) | p=0.9298 (n.s.) |
| △2016–2368 | −23.3 ± 0.9 (10) | p=0.7924 (n.s.) |
| **IV fit: Voltage-dependence of activation k$_m$ (mV)** | | |
| Wild type | 8.0 ± 0.5 (12) | p=0.9524 (n.s.) |
| Full transcript | 7.4 ± 0.5 (10) | control group |
| CKO | 12.6 ± 1.3 (11) | p<0.0001 (****) |
| △2365–2368 | 7.6 ± 0.3 (10) | p=0.9997 (n.s.) |
| △2213–2368 | 8.4 ± 0.4 (10) | p=0.7154 (n.s.) |
| △2016–2368 | 8.2 ± 0.3 (10) | p=0.8481 (n.s.) |
| **Boltzmann fit: Half-maximal activation voltage V$_{0.5}$ (mV)** | | |
| Wild type | −10.6 ± 1.4 (12) | p=0.9997 (n.s.) |
| Full transcript | −11 ± 0.1 (10) | control group |
| CKO | −1.7 ± 1.1 (11) | p<0.0001 (****) |
| △2365–2368 | −8.7 ± 1.1 (10) | p=0.6311 (n.s.) |
| △2213–2368 | −8.2 ± 1.8 (10) | p=0.4343 (n.s.) |
| △2016–2368 | −8.9 ± 1.1 (10) | p=0.7130 (n.s.) |
| **Boltzmann fit: Voltage-dependence k (mV)** | | |
| Wild type | 8.3 ± 0.5 (12) | p=0.9111 (n.s.) |
| Full transcript | 8.9 ± 0.7 (10) | control group |
| CKO | 10.3 ± 0.6 (11) | p=0.3117 (n.s.) |
| △2365–2368 | 7.7 ± 0.7 (10) | p=0.4951 (n.s.) |
| △2213–2368 | 8.9 ± 0.4 (10) | p=0.9947 (n.s.) |
| △2016–2368 | 7.2 ± 0.3 (10) | p=0.1578 (n.s.) |

*One-Way ANOVA with a Dunnett's Test with condition knockout as reference group was performed to calculate statistical significance.

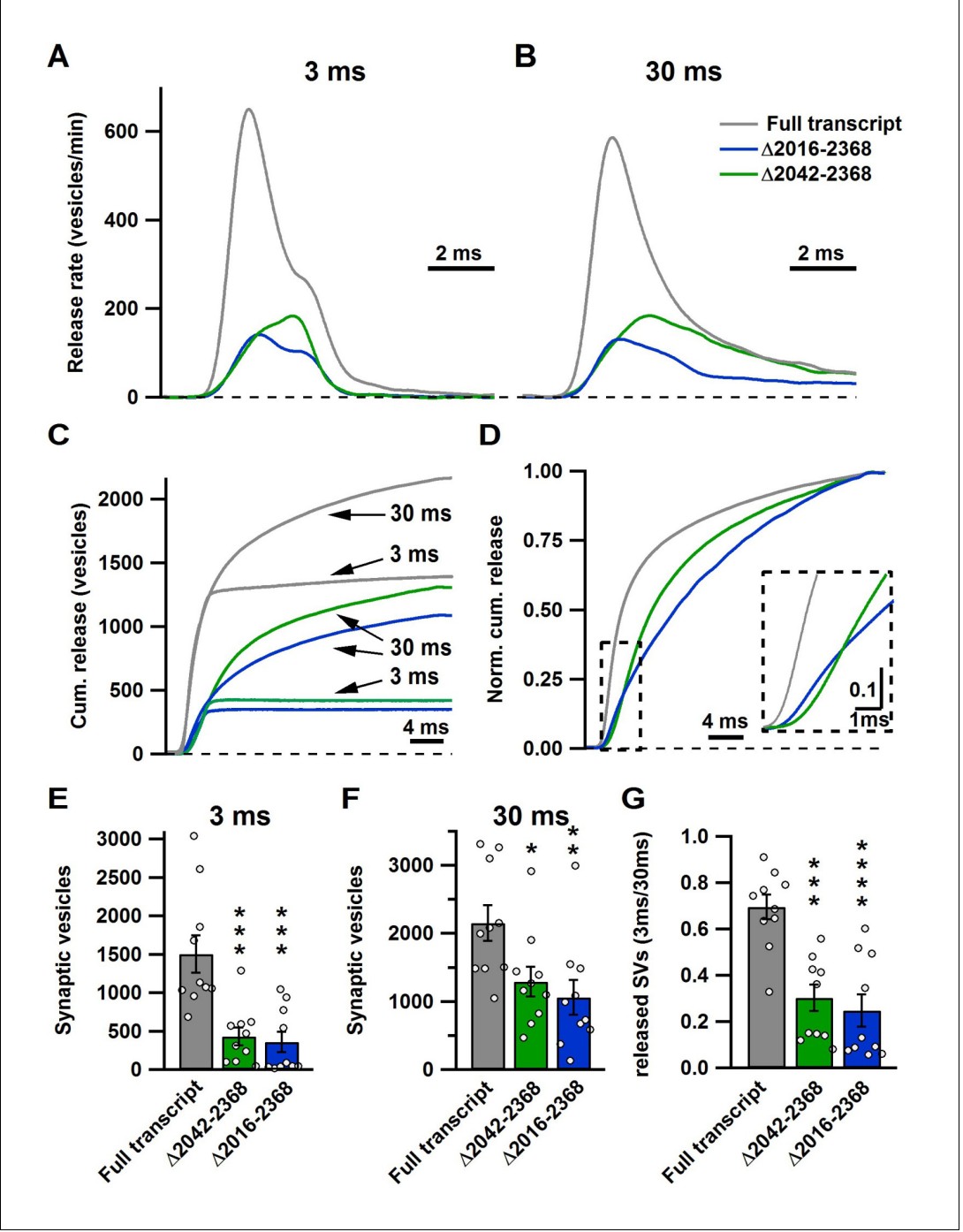

**Figure 4.** The novel C-terminal region between amino acids 2042 and 2061 regulates size of the fast and total releasable pool and synaptic vesicle release kinetics. (A–B) Average release rate trace after 3 ms (A) or 30 ms stimulation (B) from calyces expressing either Cre + full transcript rescue (grey), Δ2042–2368 (green) or Δ2016–2368 (blue); n = 10 for each group; (C) Averaged cumulative release after 3 ms and 30 ms stimulation. (D) Normalized cumulative release of the total releasable pool triggered by 30 ms stimulation. Inset presents a magnified view of the area encircled by the dashed box. (E–G) Quantification of SV numbers released by 3 ms (E) and 30 ms (F) as well as the ratio of SVs released by 3 ms and 30 ms stimulation (G). All data are depicted as mean ± SEM.

**Table 2.** Summary of currents from synaptic vesicle pool measurements.

| Parameter | 3 ms (mean ± SEM (n)) | OW-ANOVA Dunnett's Test | 30 ms (mean ± SEM (n)) | OW-ANOVA Dunnett's Test |
|---|---|---|---|---|
| **Ca²⁺ current amplitude (nA)** | | | | |
| Wild type | 0.99 ± 0.07 (10) | p=0.2671 (n.s.) | 0.93 ± 0.07 (10) | p=0.3557 (n.s.) |
| Full transcript | 1.24 ± 0.12 (10) | control group | 1.18 ± 0.12 (10) | control group |
| △2365–2368 | 1.14 ± 0.84 (10) | p=0.9320 (n.s.) | 1.01 ± 0.12 (10) | p=0.9451 (n.s.) |
| △2213–2368 | 1.04 ± 0.09 (8) | p=0.5366 (n.s.) | 0.9 ± 0.1 (8) | p=0.2766 (n.s.) |
| △2061–2368 | 1.23 ± 0.08 (10) | p=0.9999 (n.s.) | 1.13 ± 0.08 (10) | p=0.9993 (n.s.) |
| △2042–2368 | 1.13 ± 0.15 (10) | p=0.9121 (n.s.) | 1.03 ± 0.13 (10) | p=0.8410 (n.s.) |
| △2016–2368 | 0.93 ± 0.38 (10) | p=0.1091 (n.s.) | 0.86 ± 0.46 (10) | p=0.1175 (n.s.) |
| **Ca²⁺ influx charge (pC)** | | | | |
| Wild type | 2.95 ± 0.3 (10) | p=0.5713 (n.s.) | 25.93 ± 2.22 (10) | p=0.4932 (n.s.) |
| Full transcript | 3.62 ± 0.36 (10) | control group | 31.71 ± 2.87 (10) | control group |
| △2365–2368 | 3.89 ± 0.31 (10) | p=0.9838 (n.s.) | 33.52 ± 2.65 (10) | p=0.9943 (n.s.) |
| △2213–2368 | 3.48 ± 0.35 (8) | p=0.9996 (n.s.) | 28.09 ± 2.7 (8) | p=0.8924 (n.s.) |
| △2061–2368 | 3.87 ± 0.33 (10) | p=0.9893 (n.s.) | 34.94 ± 2.93 (10) | p=0.9132 (n.s.) |
| △2042–2368 | 3.8 ± 0.54 (10) | p=0.9977 (n.s.) | 33.71 ± 4.22 (10) | p=0.9910 (n.s.) |
| △2016–2368 | 3.12 ± 0.14 (10) | p=0.8063 (n.s.) | 27.81 ± 1.14 (10) | p=0.8255 (n.s.) |
| **EPSC amplitude (nA)** | | | | |
| Wild type | 8.99 ± 1.12 (10) | p=0.9995 (n.s.) | 9.77 ± 0.87 (10) | p=0.8167 (n.s.) |
| Full transcript | 8.54 ± 0.9 (10) | control group | 8.31 ± 0.79 (10) | control group |
| △2365–2368 | 6.88 ± 0.78 (10) | p=0.7024 (n.s.) | 7.45 ± 0.73 (10) | p=0.9789 (n.s.) |
| △2213–2368 | 6.05 ± 1.2 (8) | p=0.3672 (n.s.) | 6.7 ± 0.94 (8) | p=0.7880 (n.s.) |
| △2061–2368 | 7.05 ± 1.25 (10) | p=0.7836 (n.s.) | 7.72 ± 1.44 (10) | p=0.9963 (n.s.) |
| △2042–2368 | 3.38 ± 0.89 (10) | p=0.0028 (**) | 5.28 ± 1.31 (10) | p=0.1689 (n.s.) |
| △2016–2368 | 2.49 ± 0.89 (10) | p=0.0004 (***) | 3.61 ± 0.91 (10) | p=0.0098 (**) |
| **EPSC 10–90% rise time (ms)** | | | | |
| Wild type | 1.46 ± 0.14 (10) | p=0.9908 (n.s.) | 1.71 ± 0.31 (10) | p=0.9995 (n.s.) |
| Full transcript | 1.25 ± 0.13 (10) | control group | 1.41 ± 0.18 (10) | control group |
| △2365–2368 | 1.79 ± 0.09 (10) | p=0.5934 (n.s.) | 2.18 ± 0.27 (10) | p=0.9097 (n.s.) |
| △2213–2368 | 1.88 ± 0.06 (8) | p=0.4956 (n.s.) | 2.43 ± 0.3 (8) | p=0.7964 (n.s.) |
| △2061–2368 | 1.8 ± 0.12 (10) | p=0.5792 (n.s.) | 2.07 ± 0.21 (10) | p=0.9529 (n.s.) |
| △2042–2368 | 2.46 ± 0.42 (10) | p=0.0200 (*) | 4.67 ± 0.72 (10) | p=0.0046 (**) |
| △2016–2368 | 3.2 ± 0.57 (10) | p<0.0001 (****) | 6.99 ± 1.45 (10) | p<0.0001 (****) |
| **Synaptic delay (ms)** | | | | |
| Wild type | 1.82 ± 0.13 (10) | p=0.9569 (n.s.) | 1.92 ± 0.2 (10) | p=0.9977 (n.s.) |
| Full transcript | 1.7 ± 0.12 (10) | control group | 1.73 ± 0.11 (10) | control group |
| △2365–2368 | 2.11 ± 0.1 (10) | p=0.1170 (n.s.) | 2.22 ± 0.16 (10) | p=0.8464 (n.s.) |
| △2213–2368 | 2.19 ± 0.1 (8) | p=0.0615 (n.s.) | 2.33 ± 0.19 (8) | p=0.7507 (n.s.) |
| △2061–2368 | 2.14 ± 0.14 (10) | p=0.0748 (n.s.) | 2.28 ± 0.15 (10) | p=0.7714 (n.s.) |
| △2042–2368 | 2.8 ± 0.07 (10) | p<0.0001 (****) | 3.68 ± 0.3 (10) | p=0.0019 (**) |
| △2016–2368 | 2.55 ± 0.2 (10) | p<0.0001 (****) | 4.78 ± 0.83 (10) | p<0.0001 (****) |

*One-Way ANOVA with a Dunnett's Test with full transcript as a control group was performed to calculate statistical significance.

**Table 3.** Summary of 3ms /30ms EPSC ratios.

| EPSC ratio (3 ms/30 ms) | | |
|---|---|---|
| Wild type | 0.89 ± 0.05 (10) | p=0.5456 (n.s.) |
| Full transcript | 1.03 ± 0.03 (10) | control group |
| △2365–2368 | 0.92 ± 0.04 (10) | p=0.6887 (n.s.) |
| △2213–2368 | 0.87 ± 0.05 (8) | p=0.4631 (n.s.) |
| △2061–2368 | 0.93 ± 0.04 (10) | p=0.7858 (n.s.) |
| △2042–2368 | 0.61 ± 0.09 (10) | p=0.0004 (***) |
| △2016–2368 | 0.50 ± 0.12 (10) | p<0.0001 (****) |

images from the uninfected contralateral in slice control, $Ca_V2.1\Delta2042$–2368 and $Ca_V2.1\Delta2016$–2368 expressing calyces to examine whether SV docking and distribution or AZ length were altered. Analysis of EM images revealed that AZ lengths were unchanged ($Ca_V2.1\Delta2042$–2368: 267.1 ± 8.1 nm *vs.* in slice control: 280.6 ± 8.2 nm (n = 120); $Ca_V2.1\Delta2016$–2368:268. ± 7.9 nm *vs.* in slice control: 292.5 ± 8.1 nm (n = 100)), but revealed a specific reduction in only those SVs within 5 nm of the plasma membrane, in both $Ca_V2.1\Delta2042$–2368 and $Ca_V2.1\Delta2016$–2368 ($Ca_V2.1\Delta2042$–2368: 0.77 ± 0.08 nm*vs.* in-slice control: 1.39 ± 0.1 (n = 120; p<0.0001); $Ca_V2.1\Delta2016$–2368: 0.63 ± 0.08 nm *vs.* in-slice control:1.58 ± 0.12 nm (n = 100; p<0.0001) (**Figure 5**). Thus, morphological analysis revealed that the region between 2042 and 2061 in $Ca_V2.1$ $\alpha_1$ subunit regulates SV docking, and its deletion results in a reduced fast pool (RRP) size and total releasable pool size.

## Discussion

By carrying out structure function studies of the $Ca_V2.1$ $\alpha_1$ subunit on a *Cacna1a* null ($Ca_V2.1^{-/-}$) background at the calyx of Held, we were able to identify a novel intrinsic motif in the $Ca_V2.1$ $\alpha_1$ subunit's C-terminus between amino acids 2042 and 2061 that regulates SV release rates, SV docking at the AZ, and size of the fast pool (RRP for AP-evoked release), as well as the total releasable pool. Since our deletion mutants did not impact $Ca^{2+}$ currents but resulted in a slowdown in SV release rates this demonstrates that this region in the $Ca_V2.1$ $\alpha_1$ subunit plays a role in coupling SVs to $Ca_V2.1$ channels (**Figures 3**, **4** and **6**). Finally, by deleting the motifs that are proposed to directly interact with the $Ca_V2.1$ $\alpha_1$ subunit we demonstrate that the intrinsic motifs required for $Ca_V2.1$ presynaptic abundance are distinct from those responsible for coupling.

### Genetic manipulation of Ca_V2.1 at the Calyx of Held

In our study, we specifically ablated the $Ca_V2.1$ $\alpha_1$ subunit in the calyx of Held using a flox mouse line of the $Ca_V2.1$ $\alpha_1$ subunit (**Todorov et al., 2006**) which circumvents potential artifacts due to global loss of $Ca_V2.1$ in the brain and lethality issues in the *Cacna1a* KO mouse line (**Jun et al., 1999**). A major road block to studying $Ca_V2.1$ channel function in native mammalian neuronal circuits has been difficulties with the ability to make routine presynaptic molecular manipulations of $Ca_V2.1$ $\alpha_1$ subunit. The $Ca_V2.1$ $\alpha_1$ subunit (>7 kb cDNA), (**Catterall, 2011**) is larger than common viral vectors such as recombinant Adeno-associated virus (rAAV) or lentiviral vectors (rLVV), 5 kb and 9 kb maximum packaging capacity (**Lentz et al., 2012**). To overcome these challenges we utilized HdAd vectors which supplant earlier versions of recombinant Ad technology, permit packaging of up to 37 kb of foreign DNA, overcome the limitations of rAAV and rLVV, and do not impact neuronal viability (**Montesinos et al., 2016**; **Muhammad et al., 2012**; **Palmer and Ng, 2003**; **Palmer and Ng, 2005**).

We used a small modified 470 bp human synapsin promoter (hSyn) (**Kügler et al., 2003**) that is in widespread use throughout the neuroscience field. This promoter is a relatively weak promoter and does not lead to massive overexpression as seen with CMV or CBA promoters (**Glover et al., 2002**). Our rescue experiments showed $Ca_V2.1$ $\alpha_1$ subunit expression with our HdAd vectors in the $Ca_V2.1^{-/-}$ background lead to similar $Ca^{2+}$ current amplitudes, $Ca_V2.1$ subtype levels and similar SV release rates as wild-type, thereby validating our experimental approach (**Figure 1**, **Figure 2—figure supplement 1**). Thus, our HdAd vectors in conjunction with the *Cacna1a* CKO mouse line and our

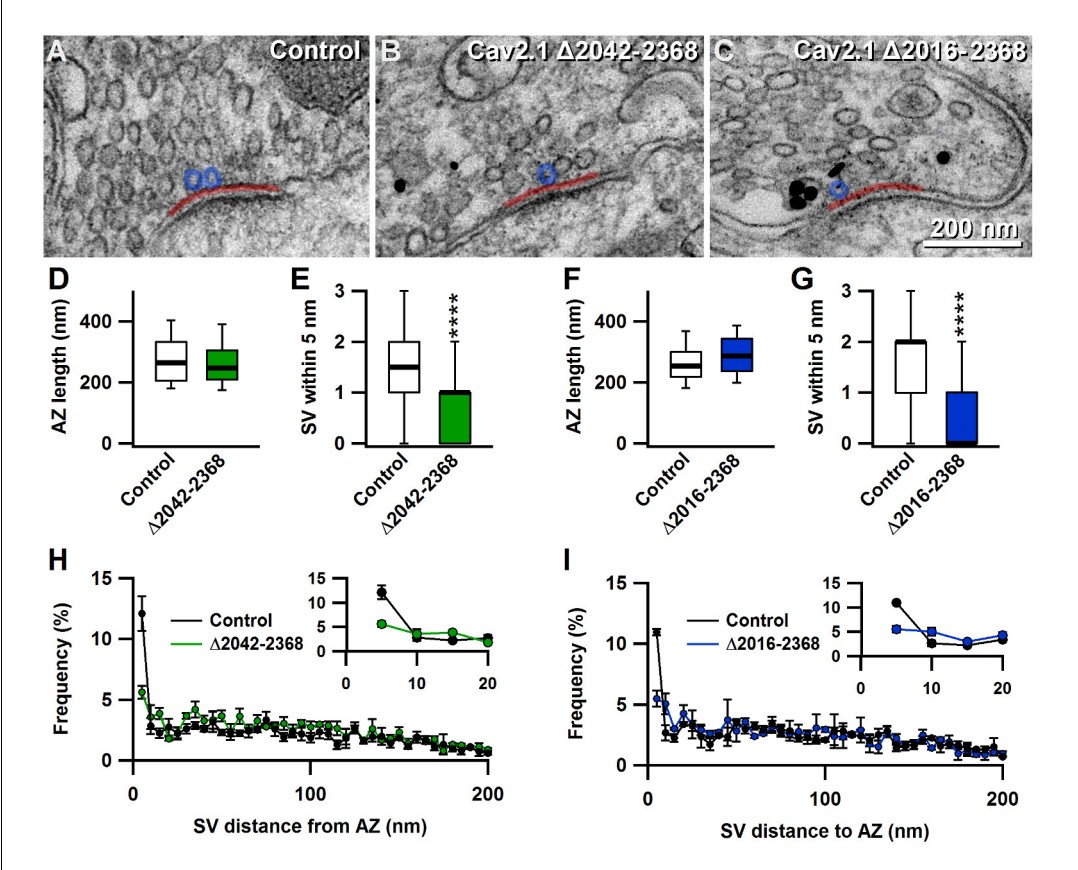

**Figure 5.** The novel C-terminal region between amino acids 2042 and 2061 regulates the number of docked synaptic vesicles at the active zones. (**A–C**) Representative EM images showing AZs from nontransduced calyces (**A**), and calyces transduced with Δ2042–2368 (**B**) or Δ2016–2368 (**C**). Transduced cells were identified by pre-embedded nanogold immunolabelling for eGFP (black dots in B and C). (**D–G**) Quantification of mean AZ length and docked SVs (within 5 nm of the membrane) of calyces transduced with Δ2042–2368 (n = 120; D E) or Δ2016–2368 (n = 100; F G), compared to AZs from the nontransduced contralateral MNTB, respectively. (**H–I**) Quantification of the mean distribution of SVs up to 200 nm distant from AZs for calyces expressing Δ2042–2368 (n = 120; H) or Δ2016–2368 (n = 100; I). Insets show SVs in closest proximity to the membrane (up to 20 nm). All data are depicted as mean ± SEM.

stereotactic surgery techniques (*Chen et al., 2013*) will be a useful platform technology to help decipher $Ca_V2.1$ function in native neuronal circuits.

## $Ca_V2.1$ localization to the presynaptic membrane

By deleting multiple AZ protein binding sites in the $Ca_V2.1$ $\alpha_1$ subunit and making direct presynaptic recordings at the calyx, we demonstrated that these binding sites and other motifs in the last 350 amino acids of the $Ca_V2.1$ $\alpha_1$ subunit are not necessary for $Ca_V2.1$ localization to the presynaptic membrane. Based on our results (*Figures 2* and *3*) which demonstrated no significant changes in $Ca^{2+}$ currents, we can rule out that proposed direct interactions with either RIM1/2 (*Kaeser et al., 2011*), MINT1, CASK (*Maximov and Bezprozvanny, 2002*; *Maximov et al., 1999*) and RBP (*Davydova et al., 2014*; *Hibino et al., 2002*) proteins are essential. Our results are similar to those studies (*Cao and Tsien, 2010*; *Hu et al., 2005*) that expressed the $Ca_V2.1$ $\alpha_1$ subunit splice variant lacking the RIM1/2, RBP, or MINT1 binding sites (*Soong et al., 2002*). This splice variant is localized to the presynaptic terminal (*Hu et al., 2005*) and could rescue the $Ca_V2.1$ channel contribution to AP evoked release in *Cacna1a* KO primary hippocampal neurons (*Cao and Tsien, 2010*). Furthermore, our data is in line with studies from *Cask* KO (*Atasoy et al., 2007*) and X11α KO (*Ho et al., 2003*) which had no impact on basal AP-evoked release kinetics and *Rim-bp1/ Rim-bp2* cKO (*Acuna et al., 2015*) animals which demonstrated no loss of $Ca_V2.1$ current density. Although our

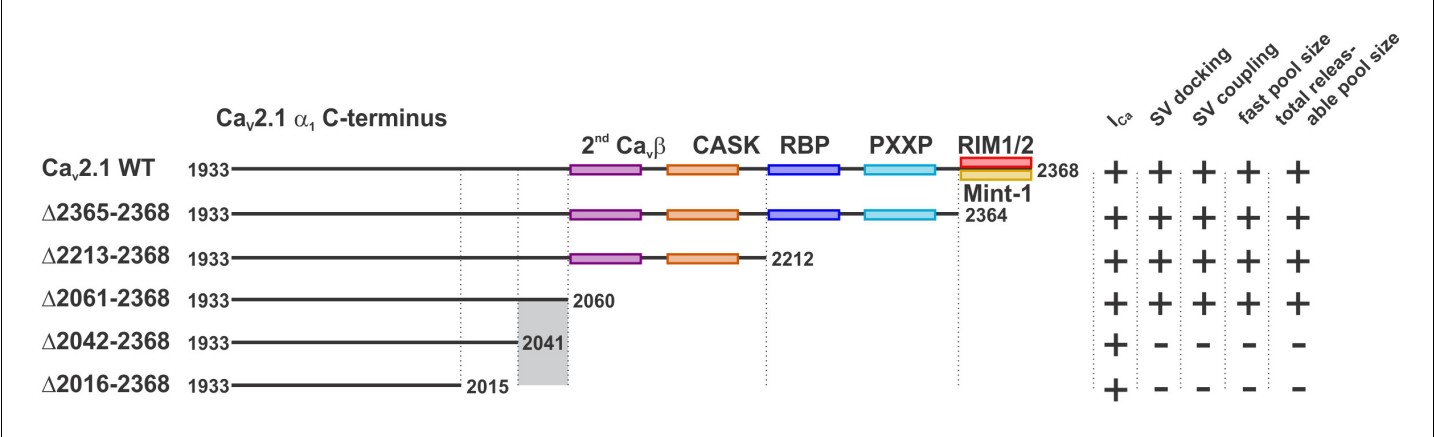

**Figure 6.** A novel C-terminal region in Ca$_V$2.1 $\alpha_1$ subunit regulates SV docking at the active zone and RRP size independent of Ca$^{2+}$ signaling. Cartoon depicting truncated regions in our Ca$_V$2.1 $\alpha_1$ deletion mutants including the binding sites for Ca$_V\beta$, CASK, RBP, RBP, RIM1/2, Mint-1 as well as PXXP motifs and the effects of truncations on I$_{Ca}$, SV docking, SV coupling as well as size of the fast and the total releasable synaptic vesicle pools. The critical region 2042–2061 is highlighted in grey.

results cannot rule out that Bassoon controls Ca$_V$2.1 abundance, our results do not support the model in which Bassoon regulates Ca$_V$2.1 abundance at the presynaptic terminal through direct RBP interaction with the Ca$_V$2.1 $\alpha_1$ subunit (*Davydova et al., 2014*).

RIM proteins have been demonstrated to be important for regulating Ca$_V$2 channel current density and abundance at both invertebrate and vertebrate presynaptic terminals, as knock out RIM proteins lead to dramatic reductions in Ca$^{2+}$ currents (*Graf et al., 2012*; *Han et al., 2011*; *Kaeser et al., 2011*). However, deletion of the DDWC motif in the Ca$_V$2.1 $\alpha_1$ subunit which interacts both with MINT1 and RIM1/2 did not lead to any changes in Ca$_V$2.1 currents (*Figures 2* and *3*) indicating this motif is not necessary for Ca$_V$2.1 targeting to the presynaptic membrane. Although the RIM1/2 PDZ domain interaction with the Ca$_V$2.1 $\alpha_1$ subunit DDWC is proposed to be critical for Ca$_V$2.1/2.2 abundance and localization to the presynaptic terminal, the necessity of this interaction was not directly demonstrated in vivo (*Kaeser et al., 2011*). Furthermore, other biochemical assays have failed to detect this direct interaction (*Wong et al., 2013*, *2014*; *Wong and Stanley, 2010*). Previous studies demonstrated that the RIM PDZ domain interacts with the CAST/ELKS proteins with an affinity of 200 nM (*Lu et al., 2005*), while the RIM PDZ domain interaction with the Ca$_V$2.1/2.2 $\alpha_1$ subunit has an affinity of 20 $\mu$M (*Kaeser et al., 2011*). Thus, an alternative interpretation is that RIM1/2 PDZ interacts with CAST/ELKS proteins to form a macromolecular complex that controls Ca$_V$2.1/2.2 abundance (*Hida and Ohtsuka, 2010*).

What then could be the motifs that are essential for Ca$_V$2.1 localization/abundance at the presynaptic terminal? In addition to the C-terminal region, the synprint region which binds to SNARE proteins has been proposed to be an integral motif for Ca$_V$2.1 incorporation into the presynaptic terminal (*Catterall, 2011*). However, syntaxin 1A binding sites in Ca$_V$2.2 are dispensable for synaptic targeting and AP-evoked release (*Szabo et al., 2006*). Interestingly, synprint domains lacking syntaxin 1A binding sites have reduced levels of incorporation into the neuroendocrine cell membranes (*Rajapaksha et al., 2008*). Another potential motif is the AID domain in the Ca$_V$2.1 $\alpha_1$ subunit which interacts with the Ca$_V\beta$ subunit to regulate Ca$_V$2.1 localization to the presynaptic membrane via a RIM1/2-dependent mechanism (*Kiyonaka et al., 2007*). Since the Ca$_V$2.1 $\alpha_1$ subunit is heavily spliced depending on the neuronal cell-type (*Simms and Zamponi, 2014*), it is possible that no single motif is responsible for Ca$_V$2.1 localization and incorporation in the presynaptic membrane, but that various motifs may act in concert or independently to ensure Ca$_V$2.1 abundance. In addition, the necessity of these motifs may also vary at different synapses and the developmental state of the neuronal circuit in which the synapses are embedded.

Ca$_V$2.1 channels are not randomly distributed in the presynaptic membrane but cluster within AZs (*Holderith et al., 2012*; *Nakamura et al., 2015*). Since the calyx of Held is a large presynaptic terminal that contains many AZs, the conclusions based our presynaptic recordings are limited to Ca$_V$2.1

localization to the presynaptic terminal. Thus, we cannot rule out that the C-terminal domains contained within amino acids 2016–2368 which are not essential for localization to the presynaptic membrane, are critical for $Ca_V2.1$ clustering/organization within individual AZs. To determine if the mechanisms that control $Ca_V2.1$ clustering and presynaptic membrane localization are independently regulated, morphological studies will need to be carried out.

Although exons 44–47 of $Ca_V2.1$ can impact voltage dependent activation and inactivation in HEK293 cells (*Hirano et al., 2017*), our direct recordings did not detect differences in the overall biophysical parameters of these mutants compared to terminals rescued with full length $Ca_V2.1$ $\alpha_1$ subunits or in a wild-type background. We did not block $Ca_V2.2$ or $Ca_V2.3$ currents which are present in prehearing calyces (*Doughty et al., 1998*; *Iwasaki and Takahashi, 1998*). Therefore, the presence of these currents could obscure possible changes in $Ca_V2.1$ current with our mutants. In addition, we did not test for these regions role in the regulation of Calcium-dependent activation or facilitation. Thus, depending on the synapse and its developmental state, it is possible that these regions are critical for modulation of $Ca_V2.1$ function in response to high frequency stimulation.

## $Ca_V2.1$ controls coupling to SVs, RRP and total releasable pool size

Although many proteins interact with the $Ca_V2.1$ channel complex (*Müller et al., 2010*), it is largely unknown whether they interact with the $Ca_V2.1$ $\alpha_1$ subunit to regulate coupling in its native environment. Our paired recordings revealed a dramatic reduction in the 3 ms peak EPSC amplitude which measures the RRP, while release kinetics with both the 3 ms and 30 ms EPSCs were dramatically slowed down in the $Ca_V2.1\Delta2042\text{-}2368$ mutant but not $\Delta2061\text{-}2368$ mutant. Thus, our data demonstrates a motif or motif(s) in amino acids 2042–2061 in the $Ca_V2.1$ $\alpha_1$ subunit regulates RRP size and coupling (*Figure 6*). Prior studies using step depolarizations at the prehearing calyx have shown that loss of RIM and RBPs resulted in a 2–3 fold slowdown in release kinetics respectively and indicating that these proteins are involved in pathways that regulate the RRP size, coupling (*Acuna et al., 2015*; *Han et al., 2011*). In contrast, we did not see any dramatic slowdown in release or changes in RRP size that mimicked these phenotypes previously seen with RIM or RBP KO animals when these direct binding motifs in the $Ca_V2.1$ $\alpha_1$ subunit were deleted. Although we observed a slight slowing of the 10–90 rise times, ~20–30%, which was similar for the $\Delta2365\text{--}2368$, $\Delta2213\text{--}2368$, $\Delta2061\text{--}2368$ mutants, this was not statistically significant. We did not directly measure AP-evoked release in this study, but it has been previously demonstrated that a $Ca_V2.1$ splice variants lacking RIM or RBP binding sites rescued AP-evoked release (*Cao and Tsien, 2010*). Finally, it has been demonstrated in PC12 cells that $Ca_V\beta$ interactions with RIM1/2 are critical for anchoring SVs to $Ca_V2$ calcium channels to control coupling (*Uriu et al., 2010*). Thus, our results strongly support that proposed individual direct interactions in between the $Ca_V2.1$ $\alpha_1$ subunit with RIM1/2 (*Han et al., 2011*; *Kaeser et al., 2011*) or RBP proteins (*Acuna et al., 2015*; *Hibino et al., 2002*) at most play a minor role in regulating RRP size and coupling.

In addition to the reduction in RRP size, we observed a reduction in the total releasable pool size, indicating a loss in the number of fusion competent vesicles. Our morphological analysis revealed that this corresponded to a 50% reduction in the number of docked SVs compared to wild-type. Recent work proposed that SV docking corresponds to priming (*Imig et al., 2014*), therefore we propose that this region in the $Ca_V2.1$ $\alpha_1$ subunit is involved in priming. Close inspection of the amino acid sequence reveals little homology to $Ca_V2.2$ and $Ca_V2.3$ $\alpha_1$ subunits (ClustalW-Gonnet series algorithm), however BLAST homology search reveals no known binding motifs (data not shown). It is possible that this region may bind to known or unknown AZ proteins that organize/cluster $Ca_V2.1$ channels in the AZ, which in turn couple to SVs and promote docking of SVs in the AZ. An alternative explanation is that removal of the 2061 to 2041 region results in a misfolding of this specific region so that $Ca_V2.1$ channels cannot interact with other proteins through other regions and cannot cluster $Ca_V2.1$ channels in the AZ. In both cases, SV docking would rely on $Ca_V2.1$ clustering through another protein to promote SV docking. However, to delineate the molecular mechanisms of how this motif regulates coupling, RRP and total releasable pool size, future experiments to identify potential binding partner(s) or solving of the $Ca_V2.1\alpha_1$ subunit structure and these mutants will need to be performed.

Despite containing $Ca_V2.1$, some presynaptic terminals transition from microdomain to nanodomain during maturation of neuronal circuits that encode temporal fidelity at high firing rates (*Baur et al., 2015*; *Fedchyshyn and Wang, 2005*). Therefore, these release states are not specific to

individual $Ca_V2$ subtypes, but instead the intrinsic motifs within the $Ca_V2.1$ $\alpha_1$ subunit are differentially utilized based on the developmental state. Our results presented here focused solely on the regulation of fast release at the prehearing calyx P9-11 which utilizes microdomain release mode (*Borst and Sakmann, 1996*; *Fedchyshyn and Wang, 2005*). Since the calyx transitions from microdomain to nanodomain release after the onset of hearing it is highly possible that intrinsic motifs in the $Ca_V2.1$ $\alpha_1$ subunit dispensable for microdomain release are necessary to support nanodomain release. Finally, since proteome composition at the AZ may vary in different various presynaptic terminals, different $Ca_V2.1$ motifs may or may not be essential to support coupling.

Taken together, these findings counter the prevailing views that (1) individual direct interactions between the $Ca_V2.1$ $\alpha_1$ subunit and RIM1/2, MINT, RBP proteins are crucial for controlling $Ca_V2.1$ abundance and coupling to SVs (*Südhof, 2013*) and (2) that PXXP motifs are involved in capturing and coupling SVs to calcium channels (*Wong et al., 2014*). Thus, our results suggest that the mechanisms of action by which known AZ proteins regulate SV coupling and $Ca_V2.1$ channel abundance involve indirect interactions with protein(s) that bind directly to the $Ca_V2.1$ $\alpha_1$ subunit.

## Materials and methods

### Animal handling and stereotactic surgery

All procedures were performed in accordance with the animal welfare laws of the Max Planck Florida Institute for Neuroscience Institutional Animals Care and Use Committee (IACUC). Stereotactic surgery was performed as described previously (*Chen et al., 2013*; *Montesinos et al., 2015*). In brief, *Cacna1a*$^{fl/fl}$ (floxed) mice (*Todorov et al., 2006*) at P1 were anesthetized by hypothermia. Subsequently, 1–2 µl HdAd (1 µl/min) in storage buffer (in mM: 10 HEPES, 250 sucrose, 1 $MgCl_2$ at pH: 7.4% and 6.6% mannitol) was injected into the aVCN using pulled glass pipettes with a 20 µm opening (Blaubrand IntraMARK, Wertheim, Germany). Two viral vectors, one expressing Cre + eGFP and the other vector one of our $Ca_V2.1$ $\alpha_1$ constructs + mCherry were co-injected (*Figure 1*). The amount of virus injected did not exceed a total of $2*10^9$ viral particles as higher amounts of viral particles have been reported to cause neuronal cell loss (*Muhammad et al., 2012*). To dissipate pressure after injection, the needle was slowly removed after the injection. After full recovery under an infrared lamp at ~37°C, pups were returned to their respective cages with their mother.

### Preparation of acute slices

Acute brainstem slices were prepared as previously described (*Chen et al., 2013*). Briefly, after decapitation of P9-P11 mice of either sex, the brains were immersed in ice-cold low $Ca^{2+}$ artificial cerebrospinal fluid (aCSF) containing (in mM): 125 NaCl, 2.5 KCl, 3 $MgCl_2$, 0.1 $CaCl_2$, 10 glucose, 25 $NaHCO_3$, 1.25 $Na_2HPO_4$, 0.4 L-ascorbic acid, 3 myo-inositol, and 2 Na-pyruvate, pH 7.3–7.4 (310 mosmol/l). Coronal 200 µm slices of the brainstem containing MNTB were obtained using a vibrating tissue slicer (Campden 7000 smz, Campden Instruments LTD, Loughborough, England) or Leica VT1200 (Leica Biosystems, Wetzlar, Germany). Slices were immediately transferred to standard aCSF (37°C, continuously bubbled with 95% $O_2$ – 5% $CO_2$) containing the same as the cutting buffer except that it contained 1 mM $MgCl_2$ and 1–2 mM $CaCl_2$. After 45 min incubation, slices were transferred to a recording chamber with the same extracellular buffer at room temperature (RT: 25°C).

### Acute slice electrophysiology

During all experiments, slices were continuously perfused with aCSF and visualized by an upright microscope (BX51WI, Olympus) through a 60x water-immersion objective (LUMPlanFL N, Olympus, Tokyo, Japan) and a CCD camera (QI-Click, QImaging, Surrey, BC, Canada) or a EMCCD camera (Luca$^{EM}$ S, Andor Technology, Belfast, UK). Patch-clamp recordings were performed by using an EPC 10/2 patch-clamp amplifier (HEKA, Lambrecht, Germany), operated by PatchMaster version 2×80 (Harvard Instruments, Holliston, MA, USA). Data were low-pass filtered at 6 kHz and sampled with a rate of 50 kHz. Calyces transduced with HdAd expressing $Ca_V2.1$ $\alpha_1$ were identified visually with two coexpressed eGFP and mCherry markers. To visualize eGFP and mCherry, slices were illuminated with light of 470 nm or 560 nm, respectively, using a Lumen 200 metal arc lamp (Prior Scientific, Rockland, MA, USA) or a Polychrome V xenon bulb monochromator (TILL Photonics, Gräfelfing, Germany).

## Presynaptic Ca$^{2+}$ current recordings

To isolate presynaptic Ca$^{2+}$ currents, aCSF was supplemented with 1 µM tetrodotoxin (TTX, Alomone labs, Jerusalem, Israel), 100 µM 4-aminopyridin (4-AP, Tocris, Bristol, UK) and 20 mM tetraethylammonium chloride (TEA, Sigma Aldrich, Darmstadt, Germany) to block Na$^+$ and K$^+$ conductance. Calyxes were whole-cell voltage-clamped at −80 mV. Current-voltage relationships were recorded in the presence of 1 mM CaCl$_2$, pharmacological isolation of VGCC subtypes was performed in 2 mM CaCl$_2$. We used 200 nM ω-agatoxin IVA (Alomone labs) to selectively block Ca$_V$2.1 and 2 µM ω-conotoxin GVIA (Alomone labs) for Ca$_V$2.2 VGCCs. Remaining current was blocked by 50 µM CdCl$_2$. All experiments to isolate Ca$_V$2 subtypes were conducted in presence of cytochrome c (0.1 mg/ml). Presynaptic patch pipettes with open tip diameters 4–6 MΩ resistance were pulled from 2.0 mm thin-walled borosilicate glass (Hilgenberg, Malsfeld, Germany) and were filled with the following (in mM): 145 Cs-gluconate, 20 TEA-Cl, 10 HEPES, 2 Na$_2$-phosphocreatine, 4 MgATP, 0.3 NaGTP, and 0.5 EGTA, pH 7.2, 325–340 mOsm). Pipettes were coated with Sylgard. Presynaptic series resistance was between 6 and 20 MΩ (usually between 10–15 MΩ) and was compensated online to 6 MΩ. Leak and capacitive currents were subtracted online with a P/5 routine. Cells with series resistance >20 MΩ and leak currents >100 pA were excluded from the analysis.

## Paired recordings

For paired recordings, calyx of Held terminals and principal neurons of MNTB were simultaneously whole-cell voltage-clamped at −80 mV and −60 mV, respectively. Patch pipettes were pulled to open tip diameters of 3.5–6 MΩ for presynaptic and to 2.5–4 MΩ for postsynaptic recordings. Both pipettes were filled with the following: (in mM): 145 Cs-gluconate, 20 TEA-Cl, 10 HEPES, 2 Na$_2$-phosphocreatine, 4 MgATP, 0.3 NaGTP, pH 7.2, 325–340 mOsm. To separate the fast and slow release components in the prehearing calyx, 0.5 mM EGTA were added in the presynaptic recording pipette (*Sakaba and Neher, 2001*). EGTA concentration in the postsynaptic pipette solution was 5 mM. Presynaptic series resistance was between 8 and 25 MΩ (usually between 10–15 MΩ) and was compensated online to 8 MΩ. Postsynaptic R$_s$ (<8 MΩ) was online compensated to R$_s$ <3 MΩ and remaining R$_s$ was further compensated offline to 0 MΩ for all EPSCs, with a custom routine (*Traynelis, 1998*) and can be found at (http://www3.mpibpc.mpg.de/groups/neher/index.php?page=software). Recordings were performed in aCSF supplemented with 1 mM MgCl$_2$ and 2 mM CaCl$_2$, cytochrome c (0.1 mg/ml; Sigma Aldrich), 100 µM 4-AP, 1 µm TTX, 50 µM D-AP5 and 20 mM TEA-Cl to isolate presynaptic Ca$^{2+}$ currents and postsynaptic AMPA receptor-mediated EPSCs. Furthermore, 2 mM kynurenic acid (Tocris) and 100 µM Cyclothiazide (CTZ, Tocris) were added to prevent saturation and desensitization of AMPA receptors and Ca$_V$2.1-mediated I$_{Ca}$ were isolated by 2 µM ω-conotoxin GVIA (Alomone labs). Cells with series resistance >20 MΩ (pre) or >10 MΩ (post) and leak currents >100 pA (pre) or >200 pA (post) were excluded from the analysis.

## Analysis of electrophysiological data

All data was analyzed offline with FitMaster version 2 × 80 (Harvard Instruments), and custom routines written in Igor Pro (version 6.37, Wavemetrics, Portland, OR, USA). Voltage dependence of channel activation was described by both peak and tail currents as functions of voltage and in FitMaster. Peak currents were fitted according to a Hodgkin-Huxley formalism with four independent gates assuming a Goldman-Hodgkin-Katz (GHK) open-channel conductance Γ:

$$I(V) = \Gamma^\star \frac{1 - e^{\frac{V - E_{rev}}{25\,mV}}}{1 - e^{-\frac{V}{25\,mV}}} \star \left(1 - e^{\frac{V - V_m}{K_m}}\right)^{-4} \tag{1}$$

with E$_{rev}$ as reversal potential, V$_m$ as half-maximal activation voltage per gate, and k$_m$ as the voltage-dependence of activation. Tail currents were measured as peaks minus baseline and fitted with a Boltzmann function:

$$I_{tail} = I_{base} + \frac{I_{min}}{1 + e^{-\frac{V - V_{1/2}}{k}}} \tag{2}$$

where V$_{1/2}$ represents the half-maximal voltage and k the corresponding slope factor.

For EPSC analysis, EPSC amplitudes were measured as peak minus baseline. Synaptic delays in response to step depolarization (step) were defined as the duration between the onset of the $I_{Ca}$ and the time at which the EPSCs were 50% of their maximum. 10–90 rise times were measured by subtracting the time at 10% of EPSC from 90% of peak amplitude. To estimate the presynaptic $I_{Ca}$ charge the presynaptic $I_{Ca}$ was integrated. The $Ca^{2+}$ charges were measured from the onset of the $Ca^{2+}$ influx to the point where 10% of the peak $I_{Ca}$ remained.

## Deconvolution

An established deconvolution approach for the calyx of Held/MNTB synapse was used to estimate quantal release rates and to measure the size of fast and slow vesicle pools (30 ms step depolarization) and fast pool contribution (3 ms step depolarization) (http://www3.mpibpc.mpg.de/groups/neher/index.php?page=software) (*Neher and Sakaba, 2001a*, *2001b*; *Sakaba and Neher, 2001*). This method compensates for residual current, caused by delayed glutamate clearance in the synaptic cleft. After subtracting the estimated residual current, it deconvolves the remaining EPSCs. We determined quantal release rates and time constants of decay by using an empirically generated template miniature EPSC (mEPSC) waveform and by further offline analysis in IgorPro (Wavemetrics). Quantal release rates were subsequently integrated to obtain the cumulative release. The fast pool was defined as the cumulative release at 3 ms. For the 30 ms long depolarization step to determine the total releasable pool, the cumulative release rates were further corrected for the refilling of the SV pools during the stimulation, assuming an average refilling rate assumed to be 10 SVs/ms.

## DNA construct and recombinant viral vector production

cDNAs, codon-optimized for expression in mouse (GeneArt, Regensburg, Germany) were used for Cre recombinase and $Ca_V2.1$ $\alpha_1$ subunit cDNA (*Mus musculus*, Accession No.: NP_031604.3). A series of mutants with deletions increasing in size from the end of the $Ca_V2.1$ $\alpha_1$ subunit cDNA C-terminal were generated to remove previously described protein interaction sites (*Butz et al., 1998*; *Davydova et al., 2014*; *Hibino et al., 2002*; *Kaeser et al., 2011*; *Maximov et al., 1999*; *Wong et al., 2013*, *2014*). Subsequently, each $Ca_V2.1$ expression cassette was cloned into the AscI site of a modified version of pdelta 28E4, gift from Dr. Philip Ng (*Palmer and Ng, 2003*) using InFusion (Clontech, Takara Bioscience, Mountain View, CA, USA). This version of pdelta28E4 has been altered by removal of 5 kb stuffer sequence and the addition of a separate neurospecific mCherry expression cassette that is driven by the 470 bp hSyn promoter. The final HdAd plasmid allows for expression of $Ca_V2.1$ independently of mCherry as a dual expression recombinant Ad vectors similar to the strategy used with second generation rAd (*Montesinos et al., 2015*, *2016*; *Young and Neher, 2009*). For HdAd Cre, the Cre recombinase cDNA was cloned into the AscI site of a different version of pdelta28E4 that has been modified to also contain a separate neurospecific EGFP expression cassette that is driven by the 470 bp hSyn promoter and the final HdAd plasmid allows for expression of Cre independently of EGFP.

Production of HdAd was carried out as previously described (*Montesinos et al., 2016*; *Palmer and Ng, 2003*; *Palmer and Ng, 2011*). Briefly, pHAD plasmid was linearized with PmeI and then transfected (Profection Mammalian Transfection System, Promega, Madison, WI, USA) into 116 producer cells, a derivative of 293N3S, developed for specifically for large scale HdAd production (*Palmer and Ng, 2003*). We did not test for mycoplasma contamintation. Helper virus (HV) was added the following day. Forty-eight hours post infection, after cytopathic effects have taken place, cells were subjected to three freeze/thaw cycles for lysis and release of the viral particles. To increase the HdAd titer, this lysate was amplified in a total of five serial coinfections of HdAd and HV from 3 × 6 cm tissue culture dishes followed by a 15 cm dish and finally 30 × 15 cm dishes of 116 cells (confluence ~90%). HdAd was purified by CsCl ultracentrifugation. HdAd was stored at −80°C in storage buffer (10 mM Hepes, 1 mM $MgCl_2$, 250 mM sucrose, pH 7.4).

## Immuno-electron microscopy

Mice (P9-11) were anesthetized with Avertin (250 mg/kg of body weight, i.p.) and perfused transcardially with warm phosphate-buffered saline (PBS, in mM: 150 NaCl, 25 $Na_2HPO_4$, 25 $NaH_2PO_4$, pH 7.4) followed by warm fixative solution for 7–9 min containing 4% paraformaldehyde (PFA), 0.5% glutaraldehyde, and 0.2% picric acid solved in phosphate buffer (PB, in mM: 100 $Na_2HPO_4$, 100

$NaH_2PO_4$, pH 7.4). Brains were postfixed with 4% PFA in PB for overnight and 50 µm coronal sections of the brainstem were obtained on a vibratome (Leica VT1200). Expression of EGFP at the calyx of Held was visualized using an epifluorescence inverted microscope (CKX41, Olympus) equipped with XCite Series 120Q lamp (Excelitas technologies, Wiesbaden, Germany) and only those samples showing EGFP were further processed as follows. After washing with PB several times, sections were cryoprotected with 10% and 20% sucrose in PB for 1 hr each, followed by 30% sucrose in PB for 2 hr and submersed into liquid nitrogen for 1 min, then thawed at room temperature. Afterwards, sections were incubated in a blocking solution containing 10% normal goat serum (NGS), 1% fish skin gelatin (FSG), 0.05% $Na_3N$ in 50 mM Tris-buffered saline (TBS, in mM: 150 NaCl, 50 Tris, pH 7.3) for 1 hr, and incubated with an anti-GFP antibody (0.1 µg/ml, ab6556, Abcam, Cambridge, UK) diluted in TBS containing 1% NGS and 0.1% FSG at 4°C for 48 hr. After washing with TBS, sections were incubated for overnight in nanogold-conjugated goat anti-rabbit IgG (1:100, Cat. No. 2003, Nanoprobes, Yaphank, NY, USA) diluted in TBS containing 1% NGS and 0.1% FSG. Immuno-labeled sections were washed in PBS, briefly fixed with 1% glutaraldehyde in PBS, washed in PBS followed by MilliQ-$H_2O$, and silver intensified for 6–8 min using HQ silver intensification kit (Nanoprobe). After washing with PB, sections were briefly rinsed with $H_2O$ and treated with 0.5% $OsO_4$ in 0.1M PB for 20 min, en-bloc stained with 1% uranyl acetate for 25 min, dehydrated in a graded series of ethanol, acetone, and propylene oxide, and flat embedded in Durcupan resin (Sigma-Aldrich). After trimming out the MNTB region, ultrathin sections were prepared with 40 nm-thickness using an ultramicrotome (EM UC7, Leica). Sections were counterstained with uranyl acetate and lead citrate, and examined in a Tecnai G2 Spirit BioTwin transmission electron microscope (FEI) at 100 kV acceleration voltage. Images were taken with a Veleta CCD camera (Olympus) operated by TIA software (FEI). Images used for quantification were taken at 60,000x magnification.

## TEM data analysis

All TEM data were analyzed using Fiji imaging analysis software (http://fiji.sc/Fiji) (*Schindelin et al., 2012*). Positive calyces were identified by the existence of gold particles, and compared to contralateral nontransduced calyces. Each presynaptic AZ was defined as the membrane directly opposing postsynaptic density, and the length of each one was measured. Vesicles within 200 nm from each AZ were manually selected and their distances relative to the AZ were calculated using a 32-bit Euclidean distance map generated from the AZ. For data analysis, vesicle distances were binned every 5 nm and counted. Vesicles less than 5 nm from the AZ were considered 'docked' (*Taschenberger et al., 2002*; *Yang et al., 2010*) and their numbers were averaged per animal. Three animals for each condition were analyzed. For both $Ca_V2.1\Delta2016$–2368 and $Ca_V2.1\Delta2042$–2368, at least 100 individual AZs were analyzed and compared with the same number of AZs of respective calyces from the contralateral MNTB (nontransduced in-slice control AZ).

## Statistical analysis

All statistical tests were conducted in Prism 6 (GraphPad Software). Sample sizes for all experiments were chosen based on assuming a population with a normal distribution, a sample size of seven is sufficient to invoke the Central Limit Theorem. All data were tested for normal distribution by performing a Shapiro-Wilk test for normality and variances of all data were estimated and compared using Bartlett's test. Electrophysiological data were compared with one-way analysis of variance (ANOVA) with a *post hoc* Dunnett's test, always using Full transcript rescue as control dataset. Patch clamp recordings lacking proper clamp quality and with high leak were excluded from data sets. EM-data were compared using an unpaired t-test. Statistical significance was accepted at $*p<0.05$; $**p<0.01$; $***p<0.001$; $****p<0.0001$. In Figures and Tables, data are reported as mean ± SEM, unless otherwise stated.

## Acknowledgements

We thank Drs. E Neher, and D DiGregorio for comments on the manuscript. We thank current and former members of the Young lab and MPFI EM core for discussion throughout the project. We thank Dr. Phillip Ng and Dr. Brendan Lee for gifts of HdAd packing plasmids and HdAd stuffer DNA, respectively. This work was supported by research grants to SMY, Jr. from the National Institutes of

Deafness and Communication Disorders (R01 DC014093) and the Max Planck Society. The authors declare no conflict of interests.

## Additional information

### Funding

| Funder | Grant reference number | Author |
|---|---|---|
| National Institute on Deafness and Other Communication Disorders | R01 DC014093 | Samuel M Young |
| Max-Planck-Gesellschaft | | Samuel M Young |

The funders had no role in study design, data collection and interpretation, or the decision to submit the work for publication.

### Author contributions

ML, ROG, Data curation, Formal analysis, Validation, Investigation, Writing—original draft, Writing—review and editing; RS, Resources, Data curation, Formal analysis, Validation, Investigation, Methodology, Writing—original draft, Writing—review and editing; TP, Data curation, Formal analysis, Validation, Investigation, Writing—review and editing; AMJMvdM, Resources, Writing—review and editing; NK, Data curation, Supervision, Validation, Methodology, Writing—original draft, Writing—review and editing; SMY, Conceptualization, Data curation, Formal analysis, Supervision, Funding acquisition, Validation, Investigation, Methodology, Writing—original draft, Project administration, Writing—review and editing

### Author ORCIDs

Samuel M Young Jr, http://orcid.org/0000-0002-7589-7612

### Ethics

Animal experimentation: All procedures were performed in accordance with the animal welfare laws of the Max Planck Florida Institute for Neuroscience Institutional Animals Care and Use Committee (IACUC) protocols (#16-001)

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
