## [Decision Letter]

Thank you for submitting your article "A novel region in the Ca_V_2.1 α_1_subunit C-terminus regulates fast vesicle fusion and vesicle docking at the active zone" for consideration by *eLife*. Your article has been reviewed by three peer reviewers, and the evaluation has been overseen by Gary Westbrook as Reviewing Editor and Senior Editor. The following individual involved in review of your submission has agreed to reveal her identity: Diane Lipscombe (Reviewer #3). The reviewers have discussed the reviews with one another and the Senior Editor has drafted this decision to help you prepare a revised submission..

Summary:

All three reviewers thought the work was on an important topic, that the experiments were a technical tour de force, and that the results will have an impact on the field. Specifically, the experiments could change how we think about the molecular function of Ca^2+^ in the synaptic vesicle release pathway in that the authors have defined a new region in CaV2.1 that appears to support vesicle release. However, there were concerns about how the new findings should be interpreted in molecular terms compared to prior work. It is very important that this be handled properly in the revision to provide honest assessment, but not overinterpretation, of the current work. In principle, rewriting/clarification should be sufficient to address the reviewers' concerns detailed below. To the extent that the authors think their work overturns dogma in the field, it is fair to state it, while acknowledging the limits of the experiments in this manuscript. Please attend to the issue with Figure 3—figure supplement 1.

Reviewer #1:

The authors have demonstrated that deletion of C-terminal region (2061–2368) of Cav2.1 channels, which interacts with active zone proteins (Rim, RBP and so on), did not impair fast transmitter release. Further deletion (2015–2060) was required to impair the coupling between Ca channels and transmitter release. In parallel, the EM analysis has shown that the number of docked vesicles was reduced. The study is interesting and may provide the evidence that single protein-protein interaction may be insufficient to explain coupling between Ca channels and readily-releasable synaptic vesicles (but see below). Also, the study is technically demanding, because they have combined paired pre- and postsynaptic recordings at the calyx of Held synapse and virus transfection in the KO mice background.

1) The authors provided no explanation about how deletion of new region in the C -terminal (2015–2060) yields loss of releasable synaptic vesicles. This region may bind to known or unknown active zone proteins to insert Ca channels in the proper place at the active zones, or else the Ca channel structure is disrupted. Without presenting alternative molecular explanations/mechanisms, the study is interesting but fails to "challenge" previous excellent studies as the authors claim at the end of ms, and confuse readers in the field. The authors should carefully discuss several possible interpretations.

2) In Table 2, it seems that release time course (ten-ninety% rise time) gets gradually slower as deletion toward N terminal proceeds. This suggests that interaction between C-terminal and active zone proteins is indeed required for fast transmitter release (I do not know understand why statistically insignificant). Please carefully deal with this issue.

Reviewer #2:

At first I thought the authors were just doing some solid, but perhaps not exciting structure-function on Cav2.1, which would be useful as not enough has been done on this front (having a 7KB cDNA is part of the reason). But, I recognize that these experiments are very challenging (in vivo rescue using a virus encoding a massive cDNA, followed by very high quality e-phys and EM). In the end, this study turned out to be much more than I had expected: it is a much needed, clear, lucid, analysis of the C-terminal region of Cav2.1, and the findings are surprising and novel. Indeed, the results indicate that much of the dogma concerning the roles played by previously reported interaction motifs is likely to be incorrect. The authors then zero-in an a novel region that is important for function. Residues 2042–2061 play a role in vesicle docking (from nice EM experiments) and in determining the size of the fast and total releasable pools of SVs. This suggest interactions that have yet to be discovered, and is a major step in our molecular understanding of how these channels mediate excitation-secretion coupling. Technically, this work looks very sound.

1) I am not clear on is the issue of how this region regulates fast release independent of abundance. The authors are asking whether 'docking' motifs for SVs and other molecular players help recruit/stabilize VGCCs in the active zone. Option one is that they do – and if so, removing those interactions should also lower the number of VGCCs in the active zone/pre-synapse that responding to depolarization. Option two is that some other element localizes VGCCs to the pre-synapse and that other element is independent of the motifs driving interactions (direct or indirect) between VGCCs and SVs. The authors found that the raw pre-synaptic Ca^2+^ currents from step depolarizations were the same for FL (they call it FT) and their truncation mutants – in subsection “Active zone protein binding sites in the CaV2.1 α1 subunit C-terminus are dispensable for CaV2.1 abundance in the presynaptic terminal” – so they conclude that the truncations did not effect pre-synaptic localization of CaV 2.1 (the Calyx had roughly the same # of VGCCs that responded to a depolarizing step to generate the same current amplitude). Of course, one caveat is that they cannot say that CaV 2.1 localization is not disrupted at the level of individual active zones – it could be possible to disrupt the localization/clustering/number of VGCCs in an active zone without effecting the overall # of VGCCs in the entire terminal (their functional measurement). I don't think they labelled CaV 2.1 in their EM images which is the experiment needed to really test this issue (and this could occur alongside or independent to the changes to SV docking that they did report). This issue should all clarified via rewriting parts of the text, and the caveat I raise above must be addressed.

Reviewer #3:

Data presented support the conclusion that amino acids 2042–2061 in mouse CaV2.1 appear necessary to support normal vesicle release from calyx of Held terminals. This region has not been previously identified and, as noted by the authors, the findings should stimulate additional research to define the role of this domain more precisely. The experiments are indeed a technical tour de force. Other claims are less well supported (points 2 and 3 below) and the interpretations need to be scaled based on data presented.

1) Coupling is defined by the authors as the "physical distance of SVs and voltage-gated calcium channels at the presynaptic terminal" (Introduction) although this is not measured in the presented studies. It would help if the authors defined this term more explicitly. The presynaptic CaV current measured by whole cell recording is used as a proxy for the abundance of CaV channels at the active zone. Is there evidence that this is valid? Could domain 2042–2061 influence clustering of CaV2.1 channels at active zones without affecting total current?

2) Figure 2 and associated supplement. CaV currents in CaV2.1 CKO terminals are reduced ~fifty% compared to WT and CaV tail currents reduced by an even smaller amount. By contrast, in Figure 1, the CaV2.1 current in presynaptic terminals is closer to eighty% of the total CaV current. Why the difference in the relative size of the CaV2.1 current between figures? Is it possible that the CKO is incomplete in experiments in Figure 2? The authors should discuss how the presence of other CaV2 currents might affect their analyses.

3) The authors conclude that CaV2.1 constructs with deletions between 2365 and 2061 have no impact on Ca current density of vesicle release. Certainly the effect size seen with 2042-2061 deletion constructs is more robust but, given the spread of the data and the relatively small population size, smaller differences between data sets might not be detectable. For example, in Figure 3. EPSC amplitudes at synapses expressing the d2213 construct (yellow) appear to be reduced relative to WT and FT samples, albeit to a smaller degree than seen in terminals expressing the d2041-2061 CaV2.1 mutants. Absent increasing population sizes to better define the distributions, the authors should be more circumspect with their conclusions. It also seems likely that molecular interactions involving the C-terminus of CaV2.1 could differ according to synapse.

4) Figure 3—figuresupplement 1. There is something wrong with this figure. The data points do not line up with the bar graphs or average values. E.g. H is missing data points; I: the WT average must be wrong; J: 3ms FT average must be wrong; K: 3ms FT average must be wrong; K: 30 ms WT average is wrong.

---

## [Author Response]

Summary:

*All three reviewers thought the work was on an important topic, that the experiments were a technical tour de force, and that the results will have an impact on the field. Specifically, the experiments could change how we think about the molecular function of Ca^2+^ in the synaptic vesicle release pathway in that the authors have defined a new region in CaV2.1 that appears to support vesicle release. However, there were concerns about how the new findings should be interpreted in molecular terms compared to prior work. It is very important that this be handled properly in the revision to provide honest assessment, but not overinterpretation, of the current work. In principle, rewriting/clarification should be sufficient to address the reviewers' concerns detailed below. To the extent that the authors think their work overturns dogma in the field, it is fair to state it, while acknowledging the limits of the experiments in this manuscript. Please attend to the issue with supplemental Figure 3.*

We thank the reviewers for their comments and criticisms. As outlined in our cover letter, *i*) We added new text to clarify how CaV2.1 localization in the presynaptic membrane could be distinct from CaV2.1 clustering in the individual active zone. *ii)* We added a new paragraph explaining how motifs that control voltage gated calcium channel to synaptic vesicle coupling may differ between synapses and developmental state of synapses in their respective neuronal circuit. *iii)* We modified text to provide a more nuanced assessment of the data rather than lead to over interpretation while acknowledging the limitations of our experiments. *iv*) We fixed supplemental Figure 3) We added absolute p values.

*Reviewer #1:*

*The authors have demonstrated that deletion of C-terminal region (2061–2368) of Cav2.1 channels, which interacts with active zone proteins (Rim, RBP and so on), did not impair fast transmitter release. Further deletion (2015–2060) was required to impair the coupling between Ca channels and transmitter release. In parallel, the EM analysis has shown that the number of docked vesicles was reduced. The study is interesting and may provide the evidence that single protein-protein interaction may be insufficient to explain coupling between Ca channels and readily-releasable synaptic vesicles (but see below). Also, the study is technically demanding, because they have combined paired pre- and postsynaptic recordings at the calyx of Held synapse and virus transfection in the KO mice background.*

*1) The authors provided no explanation about how deletion of new region in the C -terminal (2015–2060) yields loss of releasable synaptic vesicles. This region may bind to known or unknown active zone proteins to insert Ca channels in the proper place at the active zones, or else the Ca channel structure is disrupted. Without presenting alternative molecular explanations/mechanisms, the study is interesting but fails to "challenge" previous excellent studies as the authors claim at the end of ms, and confuse readers in the field. The authors should carefully discuss several possible interpretations.*

While there have been many excellent studies with genetic knock outs and biochemical data, many of these results presented in these studies are in conflict with one another. In some cases, some of these previous excellent previous studies conflict with one another are from the same group(s) as we have pointed out in our Introduction section. Therefore, we respectfully disagree that our data and results will confuse readers in the field and instead may help clarify disparate results from many different studies. Conversely, we understand the limitations of our study. Thus, we rewrote and added more text to our Discussion to present alternative molecular mechanisms for both Cav2.1 abundance and coupling that fit within the scope of a research manuscript and not a review of the field. In addition, we have toned down the conclusions as to avoid over interpretation of our results.

*2) In Table 2, it seems that release time course (ten-ninety% rise time) gets gradually slower as deletion toward N terminal proceeds. This suggests that interaction between C-terminal and active zone proteins is indeed required for fast transmitter release (I do not know understand why statistically insignificant). Please carefully deal with this issue.*

We thank the reviewer for drawing our attention to this point. Due to the multiple groups that we were comparing, we carried out an ANOVA one way test with a post hoc Dunnet’s test to compare multiple groups rather than a t-test. By carrying out the ANOVA one way test with post hoc Dunnet’s test we avoid Type I errors by carrying out multiple comparisons using a t-test. As the reviewer points out we do see a slight slowing of the rise times,~20-30%, which was similar for the Δ2365, Δ 2213, Δ 2061 mutants, while we severe slowing of the rise times with the Δ 2042 and Δ 2016 mutants compared to FT and uninjected CACNA1A flox animals. However due to the limitations of our data sample size, also brought up by reviewer#3, we don’t have the statistical power to draw a definitive conclusion that direct interactions between the regions between 2061-2368 do indeed play a role. As we pointed out in our Discussion, EPSC and SV release kinetics at the calyx of Held using step depolarizations in the RIM1/2 (Han et al. 2011) were significantly slowed down ~ 300% while in RBP knock out animals (Acuna et al. 2015) there was a slowing of~ 200% in tau fast. We would also like to point out that (Acuna et al. 2015) reported no change in AP evoked release rise times in the RBP KO background. To avoid over interpretation we have now modified our statements in the conclusions of our data in the Discussion to take into account.

We have added text in subsection”CaV2.1 controls coupling to SVs, RRP and total releasable pool size.”

“Prior studies using step depolarizations at the prehearing calyx have shown that loss of RIM and RBPs resulted in a 2-3 fold slowdown in release kinetics respectively and indicating that these proteins are involved in pathways that regulate the RRP size, coupling (Acuna et al., 2015; Han et al., 2011). In contrast, we did not see any dramatic slowdown in release or changes in RRP size that mimicked these phenotypes previously seen with RIM or RBP KO animals when these direct binding motifs in the CaV2.1 α1 subunit were deleted. Although we observed a slight slowing of the 10-90 rise times, ~20-30%, which was similar for the Δ2365-2368, Δ2213-2368, Δ2061-2368 mutants, however this was not statistically significant.. We did not directly measure AP-evoked release in this study, but it has been previously demonstrated that a CaV2.1 splice variants lacking RIM or RBP binding sites rescued AP-evoked release (Cao and Tsien, 2010). Finally, it has been demonstrated in PC12 cells that CaVβ interactions with RIM1/2 are critical for anchoring SVs to CaV2 calcium channels to control coupling (Uriu et al., 2010). Thus, our results strongly support that proposed individual direct interactions in between the CaV2.1 α1 subunit with RIM1/2 (Han et al., 2011; Kaeser et al., 2011) or RBP proteins (Acuna et al., 2015; Hibino et al., 2002) at most play a minor role in regulating RRP size and coupling.”

Also in subsection “CaV2.1 controls coupling to SVs, RRP and total releasable pool size.**”** we have removed the claim “are not essential” to now state “play at most a minor role”.

*Reviewer #2:*

*At first I thought the authors were just doing some solid, but perhaps not exciting structure-function on Cav2.1, which would be useful as not enough has been done on this front (having a 7KB cDNA is part of the reason). But, I recognize that these experiments are very challenging (in vivo rescue using a virus encoding a massive cDNA, followed by very high quality e-phys and EM). In the end, this study turned out to be much more than I had expected: it is a much needed, clear, lucid, analysis of the C-terminal region of Cav2.1, and the findings are surprising and novel. Indeed, the results indicate that much of the dogma concerning the roles played by previously reported interaction motifs is likely to be incorrect. The authors then zero-in an a novel region that is important for function. Residues 2042–2061 play a role in vesicle docking (from nice EM experiments) and in determining the size of the fast and total releasable pools of SVs. This suggest interactions that have yet to be discovered, and is a major step in our molecular understanding of how these channels mediate excitation-secretion coupling. Technically, this work looks very sound.*

*1) I am not clear on is the issue of how this region regulates fast release independent of abundance. The authors are asking whether 'docking' motifs for SVs and other molecular players help recruit/stabilize VGCCs in the active zone. Option one is that they do – and if so, removing those interactions should also lower the number of VGCCs in the active zone/pre-synapse that responding to depolarization. Option two is that some other element localizes VGCCs to the pre-synapse and that other element is independent of the motifs driving interactions (direct or indirect) between VGCCs and SVs. The authors found that the raw pre-synaptic Ca^2+^ currents from step depolarizations were the same for FL (they call it FT) and their truncation mutants in subsection “Active zone protein binding sites in the CaV2.1 α1 subunit C-terminus are dispensable for CaV2.1 abundance in the presynaptic terminal” – so they conclude that the truncations did not effect pre-synaptic localization of CaV 2.1 (the Calyx had roughly the same # of VGCCs that responded to a depolarizing step to generate the same current amplitude). Of course, one caveat is that they cannot say that CaV 2.1 localization is not disrupted at the level of individual active zones – it could be possible to disrupt the localization/clustering/number of VGCCs in an active zone without effecting the overall # of VGCCs in the entire terminal (their functional measurement). I don't think they labelled CaV 2.1 in their EM images which is the experiment needed to really test this issue (and this could occur alongside or independent to the changes to SV docking that they did report). This issue should all clarified via rewriting parts of the text, and the caveat I raise above must be addressed.*

This is an excellent point and we are in agreement as we did not label Calcium channels in the EM. We have now added a new paragraph in subsection “CaV2.1 localization to the presynaptic membrane**”**

“CaV2.1 channels are not randomly distributed in the presynaptic membrane but cluster within active zones (Holderith et al., 2012; Nakamura et al., 2015). Since the calyx of Held is a large presynaptic terminal that contains many active zones, the conclusions with our presynaptic recordings is limited to CaV2.1 localization to the presynaptic terminal. Thus, we cannot rule out that the C-terminal domains contained within amino acids 2016-2368l which are not essential for localization to the presynaptic membrane, are critical for CaV2.1 clustering/organization within individual AZs. To determine if the mechanisms that control CaV2.1 clustering and presynaptic membrane localization are independently regulated, morphological studies will need to be carried out.”

We have also added a clarification in subsection “CaV2.1 controls coupling to SVs, RRP and total releasable pool size.**”**

“….and cannot cluster Cav2.1 channels in the AZ. In both cases, SV docking would rely on Cav2.1 clustering through another protein to promote SV docking.”

*Reviewer #3:*

*Data presented support the conclusion that amino acids 2042–2061 in mouse CaV2.1 appear necessary to support normal vesicle release from calyx of Held terminals. This region has not been previously identified and, as noted by the authors, the findings should stimulate additional research to define the role of this domain more precisely. The experiments are indeed a technical tour de force. Other claims are less well supported (points 2 and 3 below) and the interpretations need to be scaled based on data presented.*

*1) Coupling is defined by the authors as the "physical distance of SVs and voltage-gated calcium channels at the presynaptic terminal" (Introduction) although this is not measured in the presented studies. It would help if the authors defined this term more explicitly. The presynaptic CaV current measured by whole cell recording is used as a proxy for the abundance of CaV channels at the active zone. Is there evidence that this is valid? Could domain 2042–2061 influence clustering of CaV2.1 channels at active zones without affecting total current?*

We have added a new paragraph in subsection”CaV2.1 controls coupling to SVs, RRP and total releasable pool size.” that explicitly defines coupling states and their impact on release modes. We have also added statements in this paragraph how the intrinsic motifs that support coupling may change at different synapses or at different developmental states.

“Despite containing Cav2.1, some presynaptic terminals transition from microdomain to nanodomain during maturation of neuronal circuits that encode temporal fidelity at high firing rates (Baur et al., 2015; Fedchyshyn and Wang, 2005). Therefore these release states are not specific to individual Cav2 subytpes, but instead the intrinsic motifs within the Cav2.1 α1 subunit are differentially utilized based on the developmental state. Our results presented here focused solely on the regulation of fast release at the prehearing calyx P9-11 which utilizes microdomain release mode (Borst and Sakmann, 1996; Fedchyshyn and Wang, 2005). Since the calyx transitions from microdomain to nanodomain release after the onset of hearing it is highly possible that intrinsic motifs in the Cav2.1 a1 subunit dispensable for microdomain release are necessary to support domain release. Finally, since proteome composition at the active zone may vary in different various presynaptic terminals, different Cav2.1 motifs may or may not be essential to support coupling.”

We explicitly defined coupling states in a new statement in the Introduction

“Differences in coupling distances between Cav2 VGCCs subtypes underpin the differences in Cav2 VGCC subtype effectiveness in eliciting AP evoked release and define the SV release mode in response to APs. (Eggermann et al., 2012). They are: nanodomain, a few tightly coupled VGCCs (<30 nm), and microdomain, many loosely coupled VGCCs (~100nm) trigger SV release (Baur et al., 2015; Eggermann et al., 2012; Fedchyshyn and Wang, 2005)”

The presynaptic CaV current measured by whole cell recording is used as a proxy for the abundance of CaV channels at the active zone. Is there evidence that this is valid? Could domain 2042-2061 influence clustering of CaV2.1 channels at active zones without affecting total current*?*

Since the calyx of Held is a large presynaptic terminal that contains many active zones, the conclusions one can make is limited to CaV localization to the presynaptic terminal. We have added a paragraph in the Discussion which outlines this issue and was addressed in response to reviewer#2.

*2) Figure 2 and associated supplement. CaV currents in CaV2.1 CKO terminals are reduced ~fifty% compared to WT and CaV tail currents reduced by an even smaller amount. By contrast, in Figure 1, the CaV2.1 current in presynaptic terminals is closer to eighty% of the total CaV current. Why the difference in the relative size of the CaV2.1 current between figures? Is it possible that the CKO is incomplete in experiments in Figure 2? The authors should discuss how the presence of other CaV2 currents might affect their analyses.*

In Figure 1, we used individual exemplary traces, while in Figure 2 since we had larger number of traces we used averaged traces to represent the data. We have now added new text to address how other CaV2 currents may affect our analysis.

This can be found in subsection “CaV2.1 localization to the presynaptic membrane”

“Although exons 44-47 of Cav2.1 can impact voltage dependent activation and inactivation in HEK293 cells (Hirano et al., 2017), our direct recordings did not detect differences in the overall biophysical parameters of these mutants compared to terminals rescued with full length CaV2.1 α1 subunits or in a wild-type background. We did not block Cav2.2 or Cav2.3 currents which are present in prehearing calyces (Doughty et al., 1998; Iwasaki and Takahashi, 1998). Therefore, the presence of these currents could obscure possible changes in Cav2.1 current with our mutants. In addition we did not test for these regions role in the regulation of calcium dependent activation or facilitation. Thus, depending on the synapse and its developmental state, it is possible that these regions are critical for modulation of CaV2.1 function in response to high frequency stimulation.”

*3) The authors conclude that CaV2.1 constructs with deletions between 2365 and 2061 have no impact on Ca current density of vesicle release. Certainly the effect size seen with 2042-2061 deletion constructs is more robust but, given the spread of the data and the relatively small population size, smaller differences between data sets might not be detectable. For example, in Figure 3. EPSC amplitudes at synapses expressing the d2213 construct (yellow) appear to be reduced relative to WT and FT samples, albeit to a smaller degree than seen in terminals expressing the d2041-2061 CaV2.1 mutants. Absent increasing population sizes to better define the distributions, the authors should be more circumspect with their conclusions. It also seems likely that molecular interactions involving the C-terminus of CaV2.1 could differ according to synapse.*

We agree with the reviewer and this point was brought up by reviewer#1. We have significantly modified text in the Discussion and made changes throughout the text to be more circumspect with our conclusions.

*4) Figure 3—figure supplement 1. There is something wrong with this figure. The data points do not line up with the bar graphs or average values. E.g. H is missing data points; I: the WT average must be wrong; J: 3ms FT average must be wrong; K: 3ms FT average must be wrong; K: 30 ms WT average is wrong.*

We apologize for this error and now have fixed Figure 3—figure supplement 1.